# High-Dimensional Bayesian Optimisation with Gaussian Process Prior Variational Autoencoders

**Siddharth Ramchandran** *
Department of Computer Science
Aalto University
Espoo, Finland

**Manuel Haussmann**
Department of Mathematics and Computer Science
University of Southern Denmark
Odense, Denmark

**Harri Lähdesmäki**
Department of Computer Science
Aalto University
Espoo, Finland

## Abstract

Bayesian optimisation (BO) using a Gaussian process (GP)-based surrogate model is a powerful tool for solving black-box optimisation problems but does not scale well to high-dimensional data. Previous works have proposed to use variational autoencoders (VAEs) to project high-dimensional data onto a low-dimensional latent space and to implement BO in the inferred latent space. In this work, we propose a conditional generative model for efficient high-dimensional BO that uses a GP surrogate model together with GP prior VAEs. A GP prior VAE extends the standard VAE by conditioning the generative and inference model on auxiliary covariates, capturing complex correlations across samples with a GP. Our model incorporates the observed target quantity values as auxiliary covariates learning a structured latent space that is better suited for the GP-based BO surrogate model. It handles partially observed auxiliary covariates using a unifying probabilistic framework and can also incorporate additional auxiliary covariates that may be available in real-world applications. We demonstrate that our method improves upon existing latent space BO methods on simulated datasets as well as on commonly used benchmarks.

## 1 Introduction

Bayesian optimisation (BO) (Mockus, 1989; Shahriari et al., 2015; Frazier, 2018) is a technique for complex optimisation problems, where the true functional form of a target quantity of interest is unknown. This target quantity may be expensive to compute or may require time consuming experiments to obtain its value. Hence, one would like to minimise the number of evaluations that are required to optimise it. Although BO offers an approach for black-box optimisation problems, it does not efficiently scale to high-dimensional data settings (Shahriari et al., 2015).

Variational autoencoders (VAEs) (Kingma & Welling, 2014; Rezende et al., 2014) are a popular family of latent-variable models that are often used to learn low-dimensional representations of high-dimensional data. The low-dimensional latent space afforded by VAEs, that is representative of the high-dimensional, potentially discrete-valued data on which it is trained, offers a powerful scaling strategy for BO. BO is performed on the inferred low-dimensional continuous-valued manifold instead of the high-dimensional data space (Gómez-Bombarelli et al., 2018). This method of combining the benefits of VAEs with BO, known as VAE BO, is a general-purpose high-dimensional black-box optimisation method with many practical applications, such as molecule discovery (Gómez-Bombarelli et al., 2018; Griffiths & Hernández-Lobato, 2020; Jin et al., 2018), neural architecture

---

*Correspondence to: `siddharth.ramchandran@aalto.fi`

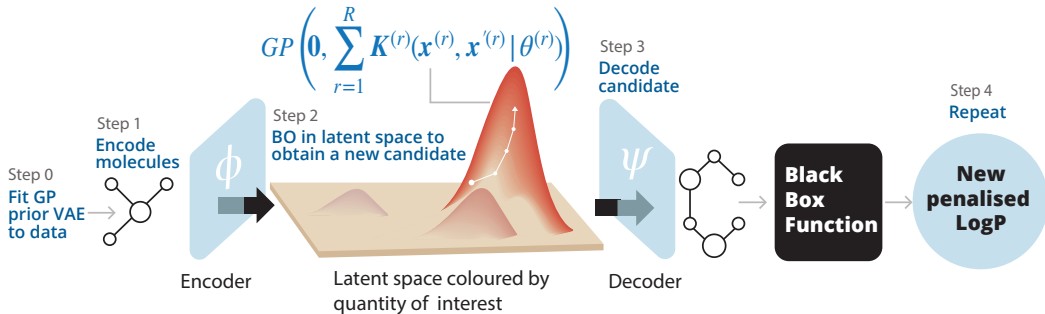

Figure 1: *An overview of our model.* Consider the example application of discovering novel drug-like molecules. Our method uses a GP prior VAE with an additive kernel over various partially observed auxiliary covariates such as molecular weight, number of hydrogen bonds, total polar surface area, etc. and the partially observed quantity of interest (represented by $x^{(r)}$ in this image for the $r^{\text{th}}$ additive kernel) to learn a structured latent space. The black-box function evaluates the quantity of interest for the chosen molecule.

search (Kandasamy et al., 2018; Ru et al., 2021) and chemical synthesis (Felton et al., 2020; Shields et al., 2021; Korovina et al., 2020).

Sohn et al. (2015) proposed conditional VAEs (cVAEs) as an extension that conditions a generative model on auxiliary covariates. However, similar to standard VAEs, this family of models ignores possible correlations between data samples. The Gaussian process (GP) prior VAE (Casale et al., 2018) extends the conditional VAE framework by replacing the i.i.d. standard Gaussian prior on the latent variables with a GP prior in order to capture arbitrary, but preferably smooth, correlations between data samples. These models have been shown to compare favourably to VAEs and cVAEs as well as effectively handle missing data in the observations. Ramchandran et al. (2024) introduced a method to impute the missing auxiliary covariates in cVAEs and thereby enhance their applicability to real-world datasets.

**Our Contribution** We propose a novel conditional deep generative model for high-dimensional BO that improves upon the existing VAE BO methods. Our proposed model uses a GP prior VAE to learn a low-dimensional, structured latent representation of the data samples, and implements the GP surrogate model to optimise the target quantity (or quantities) of interest in the repeatedly re-trained latent space. We use these partially observed target quantity values directly as auxiliary covariates to condition the GP prior VAE model. The model also incorporates additional (partially or fully) observed auxiliary covariates that may be available for a given application. Furthermore, it can effectively handle missing values in both the high-dimensional observations as well as the auxiliary covariates using a principled technique that is particularly developed for learning conditional VAEs. Fig. 1 summarises our model.

Our contributions can be summarised as follows:

- We introduce a conditional VAE-based method for efficiently performing Bayesian optimisation on high-dimensional datasets.

- We learn structured latent representations of high-dimensional data points using a GP prior VAE that handle missing values in the observations, target quantity values, and in other possible auxiliary covariates.

- We demonstrate the efficacy of our method on a synthetic dataset and on common benchmarks.

---

The source code is available at https://github.com/SidRama/GP-prior-VAE-BO.

## 2 RELATED WORKS

Bayesian optimisation is a popular black-box optimisation technique that is challenging to scale to high-dimensional data (Mockus, 1989; Shahriari et al., 2015; Frazier, 2018). Binois & Wycoff (2022) reviews the recent advancements in improving the efficiency of Bayesian Optimisation (BO) for high-dimensional problems, particularly through various structural model assumptions. To address the curse of dimensionality, Griffiths & Hernández-Lobato (2020) uses an autoencoder to learn a low-dimensional, non-linear manifold to scale BO to high-dimensional datasets. They perform a constrained BO over the latent space in order to incorporate the application-specific idiosyncrasies and thereby generate a high proportion of valid reconstructions. Stanton et al. (2022) integrate Denoising Autoencoders with a discriminative multi-task Gaussian process head into BO to learn a latent space that captures meaningful features of biological sequences. As autoencoders cannot be used to sample novel observations from their representation space, VAEs are an approach to make it possible to leverage the low-dimensional latent representation for generative purposes (Kusner et al., 2017; Gómez-Bombarelli et al., 2018). However, a vanilla VAE BO is sub-optimal as the learnt latent space is not constructed by leveraging the black-box function labels (Urtasun & Darrell, 2007; Siivola et al., 2021; Grosnit et al., 2021). Building upon this, some methods: use an automatic statistician perspective by learning the kernel combination of the surrogate GP (Lu et al., 2018), use manifold GPs in the encoder and manifold multi-output GPs in the decoder (Moriconi et al., 2020), reformulate the encoder to effectively act both as the encoder for the VAE as well as a deep kernel for the surrogate model within a local Bayesian optimisation framework using trust region method (Maus et al., 2022), and use label guidance in the latent space (Eissman et al., 2018; Tripp et al., 2020; Maus et al., 2022). Furthermore, Grosnit et al. (2021) proposed a method that combines VAEs with deep metric learning. They make use of label guidance from the labelled data points by incorporating various metric losses (e.g., triplet loss, contrasting loss, log ratio loss, etc.). However, this method does not incorporate additional information in the form of auxiliary covariates and the triplet loss requires an additional matching procedure as a pre-processing step, which can be time consuming. Other relevant works include (Notin et al., 2021; Maus et al., 2023; Lee et al., 2024)

Variational autoencoders (Kingma & Welling, 2014; Rezende et al., 2014) are popular deep learning methods that map high-dimensional, complex data to a low-dimensional space and vice-versa. Most VAE-based models assume the data to be fully observed or choose to substitute unobserved values of the encoder input with zeros (Nazabal et al., 2020; Mattei & Frellsen, 2019). Conditional variational autoencoders (Sohn et al., 2015) include information about the auxiliary covariates into both the inference and generative networks. Building upon this idea, Gaussian process prior VAEs have been proposed as an extension to incorporate arbitrary correlations as well as auxiliary covariates via Gaussian process priors (Casale et al., 2018; Fortuin et al., 2020; Ramchandran et al., 2021). These methods have shown competitive performance as well as handle missing values in the observed data. Ramchandran et al. (2024) proposed a conditional VAE-based learning approach that can robustly handle missing values in the auxiliary covariates.

## 3 BACKGROUND

Throughout the paper, we use the following notation: $\boldsymbol{y} \in \mathcal{Y}$ is a high-dimensional observation, $c \in \mathbb{R}$ is the target quantity that we want to optimise, $\boldsymbol{x} = [x_1, \dots, x_Q] \in \mathcal{X}$ denotes additional auxiliary covariates, and $\boldsymbol{z} \in \mathcal{Z} = \mathbb{R}^L$ is a $L$-dimensional latent variable. We define $\tilde{\boldsymbol{x}} = [c, \boldsymbol{x}] \in \mathbb{R} \times \mathcal{X}$. A set of $N$ observations is denoted as $Y = [\boldsymbol{y}_1, \dots, \boldsymbol{y}_N]$, with $X$, $\tilde{X}$, and $Z$ defined analogously. The target quantity $\boldsymbol{c} = [c_1, \dots, c_N]^T$ is typically partially observed.

### 3.1 BAYESIAN OPTIMISATION

Bayesian optimisation is a technique for performing efficient global optimisation of black-box functions (or unknown scoring functions) that are difficult to compute and whose functional form may not be known (Kushner, 1962; 1964; Mockus, 1989; Frazier, 2018). Given a function $f : \mathcal{Y} \mapsto \mathbb{R}$ we aim to find a point $\boldsymbol{y} \in \mathcal{Y}$ that corresponds to the global optimum of $f$. The black-box function $f$ is also referred to as a utility function as it is a measure of the target quantity, $c = f(\boldsymbol{y})$, that we are trying to optimise and informs us on the quality of the chosen sample. The problem can be written as (assuming maximisation), $\boldsymbol{y}^* = \arg\max_{\boldsymbol{y} \in \mathcal{Y}} f(\boldsymbol{y})$. Since, the unknown function $f$ is assumed to

be difficult or expensive to evaluate, Bayesian optimisation requires a surrogate model to model the true function $f$ as well as an acquisition function which is a function of the posterior and guides the process of choosing the next sample point until a stopping criteria is met or the evaluation budget $B$ is exhausted.

**Gaussian Processes and the Surrogate Model**  We use a non-parametric Gaussian process as the surrogate model of $f$ as GPs define a probability distribution over functions and for Gaussian likelihood models the posterior distribution is analytically tractable. Moreover, they maintain smoothness and uncertainty estimates to guide the exploration of new points as well as represent prior beliefs (Schulz et al., 2018). Following Williams & Rasmussen (2006), for inputs $\boldsymbol{y}, \boldsymbol{y}' \in \mathcal{Y}$, a GP is defined as $g(\boldsymbol{y}) \sim GP(\mu(\boldsymbol{y}), k(\boldsymbol{y}, \boldsymbol{y}'))$ where $\mu(\boldsymbol{y})$ is the mean and $k(\boldsymbol{y}, \boldsymbol{y}')$ is a kernel function given by $k(\boldsymbol{y}, \boldsymbol{y}') = \text{cov}(g(\boldsymbol{y}), g(\boldsymbol{y}'))$. For $N$ data points $Y = [\boldsymbol{y}_1, \ldots, \boldsymbol{y}_N]$, the induced prior probability density $g(Y) = [g(\boldsymbol{y}_1), \ldots, g(\boldsymbol{y}_N)]^T$ is a multivariate Gaussian distribution: $g(Y) \sim \mathcal{N}(\boldsymbol{0}, K_{Y,Y})$. We assume $\mu(\boldsymbol{y}) \equiv 0$ throughout this work. The elements of the covariance matrix are defined by the kernel function $[K_{Y,Y}]_{i,j} = k(\boldsymbol{y}_i, \boldsymbol{y}_j)$. GPs are intractable for large datasets as the time complexity scales by $\mathcal{O}(N^3)$. Several approximate methods have been proposed to address this through sparse Gaussian processes (Smola & Bartlett, 2000; Lawrence et al., 2002; Quinonero-Candela & Rasmussen, 2005) or via (stochastic) variational formulations (Titsias, 2009; Hensman et al., 2013) for sparse approximations.

**Acquisition Functions**  An acquisition function is a function of the posterior that captures the trade-off between exploration and exploitation of our surrogate of the function $f$ given the known evaluations. It is responsible for selecting the next candidate point in $\mathcal{Y}$ that should be evaluated or measured. We use an acquisition function $\alpha(\boldsymbol{y})$ to choose the next sample point $\boldsymbol{y}_{N+1} = \arg\max_y \alpha(\boldsymbol{y})$. A good acquisition function exploits regions around the current maximum by selecting points to query from that region while also suggesting points from unexplored regions in order to escape a local maxima. There are several candidate functions such as upper confidence bound, expected improvement, probability of improvement, and Thompson sampling (Shahriari et al., 2015). Our proposed method is agnostic to the choice of acquisition function.

## 3.2 Variational Autoencoders

We define a latent variable generative model as $p_\omega(\boldsymbol{y}, \boldsymbol{z}) = p_\psi(\boldsymbol{y} \mid \boldsymbol{z})p_\theta(\boldsymbol{z})$ which is parameterised by $\omega = \{\psi, \theta\}$, and where $\boldsymbol{z}$ is unobserved. We are generally interested in inferring this latent variable $\boldsymbol{z}$ given $\boldsymbol{y}$. The posterior distribution, $p_\omega(\boldsymbol{z} \mid \boldsymbol{y}) = p_\psi(\boldsymbol{y} \mid \boldsymbol{z})p_\theta(\boldsymbol{z})/p_\omega(\boldsymbol{y})$, is usually intractable due to the lack of a closed-form marginalisation over the latent space (Murphy, 2023). The standard VAE model comprises the generative model (the probabilistic decoder) $p_\psi(\boldsymbol{y} \mid \boldsymbol{z})$ and an inference model (the probabilistic encoder) $q_\phi(\boldsymbol{z} \mid \boldsymbol{y})$ that approximates the true posterior. VAEs use amortised variational inference that exploits the inference model $q_\phi(\boldsymbol{z} \mid \boldsymbol{y})$ to obtain approximate distributions for each $\boldsymbol{z}_n$. The encoder and decoder are typically parameterised by deep neural networks. In variational inference we minimise the Kullback-Leibler (KL) divergence from $q_\phi(\boldsymbol{z} \mid \boldsymbol{y})$ to $p_\omega(\boldsymbol{z} \mid \boldsymbol{y})$, or equivalently maximise the ELBO of the marginal log-likelihood w.r.t. $\phi$. For VAEs, approximate inference is typically conducted alongside learning the generative model's parameters, that is, w.r.t. $\phi, \psi, \theta$:

$$\log p_\omega(Y) \geq \mathcal{L}(\phi, \psi, \theta; Y) \triangleq \sum_{n=1}^{N} \mathbb{E}_{q_\phi}[\log p_\psi(\boldsymbol{y}_n \mid \boldsymbol{z}_n)] - \text{KL}[q_\phi(\boldsymbol{z}_n \mid \boldsymbol{y}_n)||p_\theta(\boldsymbol{z}_n)] \to \max_{\phi, \psi, \theta}.$$

It is straightforward to apply computationally efficient mini-batch based stochastic gradient descent to the above equation.

## 4 Our Method

### 4.1 Bayesian Optimisation with VAEs

The low-dimensional nonlinear latent manifold learnt by a VAE can be used to perform BO (Kusner et al., 2017; Gómez-Bombarelli et al., 2018; Tripp et al., 2020). The VAE is first pre-trained on the high-dimensional observations without access to the utility function values. As described in

Sec. 3.2, the encoder $q_\phi(z \mid y)$ of the learnt VAE is used to map the observations $y \in \mathcal{Y}$ onto a low-dimensional latent representation $z \in \mathcal{Z}$. The VAE-based methods then perform latent space optimisation (LSO) (Tripp et al., 2020) by fitting a surrogate model over the latent space to model the utility function of interest. The VAE BO aims to identify a $z^*$ such that the corresponding $y^*$, that is obtained from the pre-trained decoder, minimises a utility function of interest, $f(y^*)$. In other words, we would like to obtain a $z^*$ such that we maximise the expectation over the utility function evaluated on $y^* \sim p_\psi(y^* \mid z^*)$, i.e., $\arg\max_{z \in \mathcal{Z}} \mathbb{E}_{y \sim p_\psi(\cdot \mid z)}[f(y)]$. Once we have a new $y^*$ and its associated utility function value $c$, we append them to the training dataset and update the parameters $\phi$ and $\psi$ either after each BO step or at a chosen frequency. Tripp et al. (2020) use this approach with the help of a weighted retraining scheme according to their utility function values.

## 4.2 GAUSSIAN PROCESS PRIOR VAEs FOR BO

A limitation of standard VAE BO is that it infers an unconditional latent-variable model without any guidance from the observed target quantities. Departures from this limitation have been proposed, e.g., in (Eissman et al., 2018; Tripp et al., 2020; Maus et al., 2022). Recently, Grosnit et al. (2021) built upon VAE BOs by using deep metric learning to actively steer the generative model to maintain a latent manifold that is useful for the BO task. We propose to use GP prior VAEs that guide the generative model by conditioning the GP prior with auxiliary covariates.

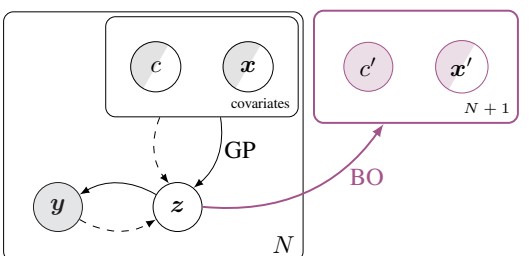

Figure 2: *Our proposed model.* Solid lines refer to the generative model and dashed lines to the inference model. Empty circles are unobserved, shaded circles are observed, and partially shaded circles are partially observed. Target quantity $c'$ and possible additional covariates $x'$ refer to the new candidate observation that will be added to the training set.

The key distinction of GP prior VAEs is that the factorisable conditional prior defined over the latent space $p_\theta(Z|X) = \prod_{i=1}^{N} p_\theta(z_i \mid x_i)$ is replaced by a GP prior. Assuming a function $\tau : \mathcal{X} \to \mathcal{Z}$, which maps auxiliary covariates to the $L$-dimensional latent space, we denote $z = \tau(x) = (\tau_1(x), \ldots, \tau_L(x))^T$. GP prior VAEs model each latent dimension with an independent GP $\tau_l(x) \sim \mathcal{GP}(\mu_l(x), k_l(x, x' \mid \theta_l))$, where $\mu_l(x)$ is the mean, $k_l(x, x' \mid \theta_l)$ is the covariance function, and $\theta_l$ denotes the parameters of the covariance function. The GP prior for the $l^{\text{th}}$ latent dimension can be written as a joint multivariate Gaussian distribution for the function values $\bar{z}_l = \tau_l(X) = (\tau_l(x_1), \ldots, \tau_l(x_N))^T$, such that $p_\theta(\bar{z}_l \mid X) = p_\theta(\tau_l(X)) = \mathcal{N}(\bar{z}_l \mid \mathbf{0}, K_{XX}^{(l)})$, where $\{K_{XX}^{(l)}\}_{i,j} = k_l(x_i, x_j \mid \theta_l)$. Our joint conditional prior is $p_\theta(Z \mid X) = \prod_{l=1}^{L} p_\theta(\bar{z}_l \mid X) = \prod_{l=1}^{L} \mathcal{N}(\bar{z}_l \mid \mathbf{0}, K_{XX}^{(l)})$.

We propose to learn a low-dimensional latent embedding for BO using a GP prior VAE that is conditioned on the target quantity of interest, i.e., $p_\theta(Z \mid c)$. We hypothesise that using the target quantity as the conditioning variable will automatically guide the latent embeddings to a smooth manifold that is beneficial for the BO task. Since the target quantity $c \in \mathbb{R}$, the GP prior VAE can be defined using any of the commonly used smooth kernel functions, such as the squared exponential kernel. Following the same reasoning, if data points $y$ have any additional known properties $x$, we can incorporate those in the GP prior VAE framework as well by conditioning the latent variable generation with both $c$ and $x$ (we denote $\tilde{x} = [c, x]$), i.e., $p_\theta(Z \mid \tilde{X})$. If all auxiliary covariates in $\tilde{x}$ are continuous, we could incorporate $\tilde{x}$, e.g., via a single squared exponential kernel with a shared length-scale parameter or use an automatic relevance determination (ARD) kernel to define covariate-specific length-scales. In practice, however, some of the auxiliary covariates may be, e.g., binary or categorical. Ramchandran et al. (2021) have shown that it is possible to have flexible and expressive covariance functions depending on the nature of the auxiliary covariates. In this work, we similarly assume $Q + 1$ additive covariance functions, $k_l(\tilde{x}, \tilde{x}' \mid \theta_l) = k_l(c, c' \mid \theta_l) + \sum_{r=1}^{Q} k_{l,r}(x_r, x'_r \mid \theta_{l,r}) + \sigma_{zl}^2$, implying that $K_{\tilde{X}\tilde{X}}^{(l)} = K_{cc}^{(l)} + \sum_{r=1}^{Q} K_{X_r X_r}^{(l,r)} + \sigma_{zl}^2 I_N$, where the choice of the kernels depends on the application and $X_r$ denotes the $r^{\text{th}}$ auxiliary variable.

---

**Algorithm 1:** An overview of our proposed algorithm

---

**Input:** Budget $B$, frequency $\nu$, initial dataset $\mathcal{D}$, pre-trained VAE

**for** $j = 1$ **to** $J \equiv \lceil B/\nu \rceil$ **do**
 // Train the GP prior VAE on $\mathcal{D} = \mathcal{D}_{\mathbb{O}} \cup \mathcal{D}_{\mathbb{U}}$
 Solve $\phi_j^*, \psi_j^*, \theta_j^* \leftarrow \arg\max_{\phi,\psi,\theta} \mathrm{ELBO}_{\text{GP-VAE-miss}(\phi,\psi,\theta)}[\mathcal{D}]$;
 Compute $\mathcal{D}_{\mathbb{Z}} \leftarrow \langle \boldsymbol{z}_i, f(\boldsymbol{y}_i) \rangle_{i \in \mathcal{I}_{\mathbb{O}}}$ by using the encoder $\phi_j^*$ to obtain $\boldsymbol{z}_i$ ;
 **for** $k = 0$ **to** $\nu - 1$ **and** $EI(\hat{\boldsymbol{z}}_{j,k+1}) \geq \eta$ **do**  // Perform Bayesian optimisation
  Fit surrogate GP on $\langle \boldsymbol{z}_i, f(\boldsymbol{y}_i) \rangle_{i \in \mathcal{I}_{\mathbb{O}}}$;
  Optimise $EI$ for $\hat{\boldsymbol{z}}_{j,k+1}$;
  Use decoder $\psi_j^*$ to map $\hat{\boldsymbol{z}}_{j,k+1}$ to $\hat{\boldsymbol{y}}$;  // Decode point chosen by B.O.
  Evaluate $c = f(\hat{\boldsymbol{y}})$, augment data $\mathcal{D}_{\mathbb{O}}, \mathcal{D}_{\mathbb{Z}}$;  // Evaluate black-box function
  Increment $N^o$;
 **end**
**end**

**Output:** $\boldsymbol{y}^* = \arg\max_{\boldsymbol{y} \in \mathcal{D}_{\mathbb{O}}} f(\boldsymbol{y})$

---

## 4.3 Partially Observed Target Quantity and Additional Covariates

In the BO setting, the target quantity of interest that we are optimising is typically available only for a (very) small number of data points. This is problematic for conditional generative models, such as GP prior VAEs, as they assume that covariates that are used to condition the generation are always known and observed. Moreover, in our problem setting, the additional auxiliary covariates that may be available in a specific application may also have missing values. We follow a formulation similar to that of Ramchandran et al. (2024) to handle the missing values in the covariates.

We augment our generative model with a prior distribution, $p_\lambda(\tilde{\boldsymbol{x}})$, factorising over $\tilde{\boldsymbol{x}}$, parameterised by $\lambda$. Representing the observed and unobserved parts as $Y = (Y^{\mathrm{o}}, Y^{\mathrm{u}})$ and $\tilde{X} = (\tilde{X}^{\mathrm{o}}, \tilde{X}^{\mathrm{u}})$, we approximate the true posterior distribution of the unobserved variables $Z$ and $\tilde{X}^{\mathrm{u}}$, represented as $p_\gamma(Z, \tilde{X}^{\mathrm{u}} \mid Y^{\mathrm{o}}, \tilde{X}^{\mathrm{o}})$ and parameterised by $\gamma = \{\psi, \theta, \lambda\}$, using amortised variational inference. We make use of a conditionally independent, factorisable variational approximation: $q_\phi(Z, \tilde{X}^{\mathrm{u}} \mid Y^{\mathrm{o}}, \tilde{X}^{\mathrm{o}}) = q_\phi(Z \mid Y^{\mathrm{o}}, \tilde{X}^{\mathrm{o}}) q_\phi(\tilde{X}^{\mathrm{u}} \mid \tilde{X}^{\mathrm{o}}) = \prod_{i=1}^N q_\phi(\boldsymbol{z}_i \mid \boldsymbol{y}_i^{\mathrm{o}}, \tilde{\boldsymbol{x}}_i^{\mathrm{o}}) q_\phi(\tilde{\boldsymbol{x}}_i^{\mathrm{u}} \mid \tilde{\boldsymbol{x}}_i^{\mathrm{o}})$. The latent variables $\boldsymbol{z}_i$ are assumed to have a Gaussian variational distribution and, for the discrete and continuous-valued covariates $\tilde{\boldsymbol{x}}_i^{\mathrm{u}}$, categorical and Gaussian distributions respectively. Following Ramchandran et al. (2024), we write the ELBO objective with missing covariates ($\mathrm{ELBO}_{\text{GP-VAE-miss}}$) as

$$\log p_\gamma(Y^{\mathrm{o}} \mid \tilde{X}^{\mathrm{o}}) \geq \mathbb{E}_{q_\phi}[\log p_\psi(Y^{\mathrm{o}} \mid Z)] - \mathbb{E}_{q_\phi}\left[\mathrm{KL}[q_\phi(Z \mid Y^{\mathrm{o}}, \tilde{X}^{\mathrm{o}}) || p_\theta(Z \mid \tilde{X}^{\mathrm{u}}, \tilde{X}^{\mathrm{o}})]\right] \quad (1)$$
$$- \mathrm{KL}[q_\phi(\tilde{X}^{\mathrm{u}} \mid \tilde{X}^{\mathrm{o}}) || p_\lambda(\tilde{X}^{\mathrm{u}} \mid \tilde{X}^{\mathrm{o}})],$$

where the first and the second expectations are with respect to the latent variables $Z$ and missing covariates $\tilde{X}^{\mathrm{u}}$, respectively, and can be approximated using Monte Carlo (see the Sec. A of the Appendices for details of deriving the ELBO). For each specific value of the missing covariates, the KL divergence in Eq. 1 has a computation complexity of $\mathcal{O}(N^3)$. Earlier work by Ramchandran et al. (2021; 2024) has shown that using the low-rank inducing point approximation for the multi-output GP $p_\theta(Z \mid \tilde{X}^{\mathrm{u}}, \tilde{X}^{\mathrm{o}}) = p_\theta(Z \mid X)$, one can derive a scalable ELBO that provides an unbiased, mini-batch compatible lower bound for efficient learning. See the Sec. A.1 of the Appendices for the specific expression of the scalable lower bound that we use.

## 4.4 High-Dimensional BO with Gaussian Process Prior VAEs

We use the latent space learnt by the GP prior VAE to perform efficient BO. In particular, our method can handle missing values in both the observations $\boldsymbol{y}$ and covariates $\boldsymbol{x}$ (partially observed features denoted as $\boldsymbol{y}^{\mathrm{o}}$ and $\boldsymbol{x}^{\mathrm{o}}$), as well as large datasets through the scalable ELBO described in Sec. 4.3. Consider a dataset $\mathcal{D} = \mathcal{D}_{\mathbb{O}} \cup \mathcal{D}_{\mathbb{U}}$ where $\mathcal{D}_{\mathbb{O}}$ represents the data points whose target quantity $c$ is observed (indexed by $\mathcal{I}_{\mathbb{O}}$) and $\mathcal{D}_{\mathbb{U}}$ represents the data points whose $c$ is unobserved (indexed by $\mathcal{I}_{\mathbb{U}}$). Therefore, $\mathcal{D}_{\mathbb{O}} = \{\boldsymbol{y}_i^{\mathrm{o}}, \boldsymbol{x}_i^{\mathrm{o}}, c_i\}_{i \in \mathcal{I}_{\mathbb{O}}}$, with $c_i = f(\boldsymbol{y}_i)$ and $\mathcal{D}_{\mathbb{U}} = \{\boldsymbol{y}_i^{\mathrm{o}}, \boldsymbol{x}_i^{\mathrm{o}}\}_{i \in \mathcal{I}_{\mathbb{U}}}$, where, as before,

$\boldsymbol{x}_i \in \mathcal{X}$ refers to the additional auxiliary covariates (that may or may not be available, depending on the application), and $f(\cdot)$ refers to an expensive black-box function. $N$ refers to the total number of observations, comprising the number of observations with observed quantity of interest $N^o = |\mathcal{I}_{\mathbb{O}}|$ and number of observations with unobserved quantity of interest $N^u = |\mathcal{I}_{\mathbb{U}}|$.

**Algorithm** See Algorithm 1 for a pseudo-code summary of our method. Budget $B$ refers to the maximum number of evaluations of the black-box function that can be performed, $\nu$ refers to the number of BO steps performed before re-optimising our GP prior VAE model with the augmented dataset $\mathcal{D}$, and $EI$ pertains to the expected improvement acquisition function (the algorithm is agnostic to this choice).

We obtain the optimal encoder and decoder parameters ($\phi_l^*$ and $\psi_l^*$ respectively) by optimising the ELBO (in Sec. 4.3). The method computes the fully-observed, low-dimensional latent space representation $\boldsymbol{z}_i$ of the observations $\boldsymbol{y}_i^o$ using the optimal encoder at the current iteration, and implements a BO step. The new chosen observation $\hat{\boldsymbol{y}}$, its covariates (if known), and the obtained target quantity of interest $c = f(\hat{\boldsymbol{y}})$ are appended to $\mathcal{D}_{\mathbb{O}}$. After budget has been exhausted, our algorithm returns the best candidate acquired so far.

We periodically re-train and fit a conditional generative model using the entire dataset — comprising both the initial data and samples collected during the BO steps, where covariates may be partially observed. Unlike Tripp et al. (2020), our model fitting remains unbiased toward high objective values. Instead, periodic training guides the embeddings of high-dimensional samples toward a smooth manifold, as specified by the GP prior, which conditions on both the objective values and any available auxiliary covariates. The BO algorithm operates in this learned latent space, inherently structured for the BO surrogate model. Ultimately, it is the BO and its acquisition function that drives the preference for higher objective values, as in classical BO.

## 5 Experiments

We demonstrate the efficacy of our method described in Algorithm 1 on simulated datasets as well as on a molecular discovery benchmark dataset. In all our experiments, $10\%$ of the training data is used as a held-out validation set for early-stopping to ensure that the generative model does not overfit. We describe the neural network architectures in the Sec. F of the Appendices. We use the same BO options and underlying VAE architectures for all methods in a particular experiment. We benchmark against the following methods:

**LSO** This latent space optimisation method uses a VAE to learn a low-dimensional representation of the high-dimensional dataset. BO is performed over the low-dimensional latent space. LSO does not take into account any auxiliary covariate information and makes use of a standard Gaussian prior over the latent space.

Grosnit et al. (2021) proposed a VAE-based method that tries to construct discriminative latent spaces for VAE-based BO methods by incorporating a metric loss term in the ELBO. We compare our model against the triplet loss, log-ratio loss, and contrastive loss.

**Triplet Loss (T-LBO)** As described in Grosnit et al. (2021), the triplet loss measures distances between input triplets. In other words, this loss tries to introduce a structured space where positive and negative pairs cluster together subject to separation by a margin. The triplet pairs are assigned as a pre-processing step.

**Log-Ratio Loss (LR-LBO)** This metric loss is described in Kim et al. (2019) and is a continuous metric loss that is applied to triplets of inputs. This loss is used with the model described in Grosnit et al. (2021).

**Contrastive Loss (C-LBO)** This deep metric loss is described in Hadsell et al. (2006). The contrastive loss operates on input pairs by separating the latent encodings based on class label information. This is used with the model described in Grosnit et al. (2021).

**Local Latent Bayesian Optimisation (LOL-BO)** As proposed in Maus et al. (2022), this method is a latent space BO approach that addresses the mismatch between the notion of a trust region in the latent space and a trust region in the structured input space.

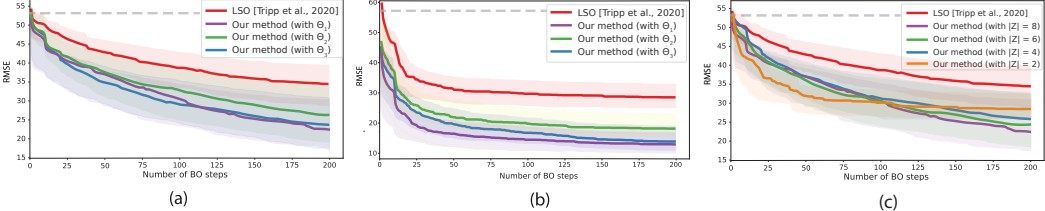

Figure 3: *Results from our experiments with a synthetic dataset.* **Lower values are better**. (a) Comparing the performance of our model with the LSO benchmark. The dataset comprises $500$ instances out of which the target quantity is observed only for $100$ instances. $\Theta_1$ pertains to an additive GP prior VAE over all three covariates $\boldsymbol{x}$ and the partially observed quantity of interest $c$, $\Theta_2$ to a GP prior VAE over only the partially observed target $c$, and $\Theta_3$ to an additive GP prior VAE over the partially observed quantity of interest $c$ and shift$_\text{x}$. (b) Similarly, we also demonstrate our model's performance on a dataset which comprises $5000$ instances out of which the quantity of interest is observed only for $500$ instances. (c) Effect of the choice of latent dimension with the dataset comprising $500$ instances. $|Z|$ pertains to the dimensionality of the latent space. All plots depict the mean quantity of interest value with the $95\%$ confidence interval (shaded region) obtained over $100$ repetitions with regenerated training data and target images. The grey line pertains to the lowest RMSE in the training set.

## 5.1 DEMONSTRATION ON SYNTHETIC DATA

We demonstrate our model's ability to perform effective high-dimensional BO by modifying digits from the MNIST dataset. In particular, we randomly select an instance of a digit and resize this digit to a dimension of $52 \times 52$ pixels for a larger image space. We perform three different manipulations (which would form our additional auxiliary covariates $\boldsymbol{x}$) to this digit: rotation about the centre, shift along the $x$-axis, and shift along the $y$-axis, by stochastically choosing these values $500$ or $5000$ times. Our final training set comprises either $500$ or $5000$ samples. Furthermore, we define a *'target image'*, $\boldsymbol{y}^{\text{target}}$, with a particular rotation and shift values. This target image is not included in the training set and we ensure that the manipulations are sufficiently different from the target values. See Suppl. Fig. 7 for a random sample of the training data.

In this experiment, the black-box function is the root mean squared error (RMSE) between the unseen target digit and a chosen digit (either a digit from the training set or a new candidate), i.e., $f(\boldsymbol{y}) = \text{RMSE}(\boldsymbol{y}^{\text{target}} - \boldsymbol{y})$. Our objective is to find a new digit that minimises the value returned by the black-box function. In other words, we want the quantity of interest to be as close to zero as possible. Furthermore, in our training set, we assume that the quantity of interest $c = f(\boldsymbol{y})$ (i.e. the RMSE between the unseen target and the digit) is known for only a few digits (i.e., $|\mathcal{I}_{\mathbb{O}}|$ equals $100$ or $500$) and unobserved for the rest.

To ensure that there is sufficient stochasticity in the choice of targets, we repeat our experiments as well as the generation of data $100$ times. We fit our GP prior VAE-based method by making use of the partially observed target quantity of interest $c$ as well as (a subset of) the auxiliary covariates $\boldsymbol{x}$. We make use of Algorithm 1 with $\nu = 10$ as well as $B = 200$. We experiment with different choices of kernels to empirically obtain the optimal model.

Fig. 3 demonstrates our experiments on the synthetic dataset and we can see that our method finds candidate points with a quantity of interest (or RMSE) that are significantly lower than those in the training set. In Fig. 3(a) we demonstrate the performance of our model on $500$ instances out of which the quantity of interest is observed for $100$ instances. The LSO method is trained with all $500$ instances and the BO computation is performed using the $100$ instances for which the quantity of interest is observed. Our method outperforms the LSO model already when the GP prior VAE is fitted with the partially observed target quantity ($\Theta_2$), and results improves further if additional auxiliary covariates are available ($\Theta_1$ and $\Theta_3$).

Similarly, Fig. 3(b) demonstrates that our method outperforms the baseline LSO with $5000$ instances out of which the quantity of interest is observed for $500$ instances and Fig. 3(c) demonstrates the effect of the choice of latent dimension.

We visualise the learnt latent space and the BO steps taken in Suppl. Fig. 10. Our method learns latent representations where the target quantity increases smoothly from the lower left corner to the upper right corner (though we again emphasise that this is a 2-D UMAP (McInnes et al., 2018) visualisation). The BO steps explore the region of the latent space where the target quantity has high values. In Suppl. Fig. 6, we perform ablations with different subsets of additional auxiliary covariates. Furthermore, we demonstrate the performance of Vanilla VAE BO in Suppl. Fig. 5.

## 5.2 Expression Reconstruction

We consider the common task of generating single-variable mathematical expressions from a formal grammar (Kusner et al., 2017; Tripp et al., 2020; Grosnit et al., 2021; Maus et al., 2022). The objective is to minimise a distance/regret based on mean squared error (MSE) (defined as $\log(1 + \text{MSE})$) between a generated expression and the target expression, $x * \sin(x * x)$. We followed the data preparation proposed by Grosnit et al. (2021) to obtain $40000$ data points and augmented the data with 8 additional covariates (count of the elements '/', '*', '+', 'exp', 'sine', '1', '2', and '3' in the expressions) which can be easily gleaned from the expressions. In order to appropriately handle the mathematical expressions, we use the Grammar VAE (Kusner et al., 2017). To demonstrate the efficacy of our method, we make use of Algorithm 1 with $\nu = 10$ and $B = 500$ as well as an additive kernel over the 8 additional covariates and regret. Fig. 4(a) demonstrates that our method achieves competitive performance against the benchmark methods. In Suppl. Fig. 11, we visualise the mean regret achieved by our method together with the $95\%$ confidence interval.

## 5.3 Molecule Optimisation

We use the ZINC-250K molecular dataset used in Gómez-Bombarelli et al. (2018), which consists of $250000$ drug-like commercially available molecules extracted from the ZINC database (Irwin et al., 2012) - a public dataset for ligand discovery. The dataset includes the molecular structures in the SMILES string representation (Weininger, 1988) and three molecular properties: the water-octanol partition coefficient (logP), the Synthetic Accessibility Score (SAS), and the Quantitative Estimation of Drug-likeness (QED) (Bickerton et al., 2012). The objective of the task is to maximise the penalised logP which is defined as the logP penalised by the SAS and the number of long cycles: $\text{penalised logP}(m) = \text{logP}(m) - \text{SAS}(m) - \text{cycle}(m)$ where $m$ is the molecular instance and $\text{cycle}(\cdot)$ is the number of long cycles.

We augmented the ZINC-250K with five additional covariates: molecular weight, number of hydrogen donors, number of hydrogen acceptors, number of rotatable bonds, and total polar surface area. These values were computed using a popular open-source chem-informatics tool, RDKit (Landrum et al., 2013) (see Suppl. Fig. 12 for a visualisation of the distribution of these properties in the form of histograms). Including QED and SAS, there are seven additional auxiliary covariates and the penalised logP is the quantity of interest which we are trying to maximise. For a new molecule, it is possible to compute the penalised logP using RDKit (acting as our black-box function). Furthermore, we assume that the penalised logP is partially observed (observed for only $1\%$ of the data).

We demonstrate the ability of our model to optimise the structure of the molecule in order to maximise a property of interest (the penalised logP). To handle the SMILES representation of the molecules, we use the Junction Tree VAE (JT-VAE) (Jin et al., 2018), which introduced an encoder and decoder suitable to molecular graphs. In our experiments, we extend the implementation by Grosnit et al. (2021). We note that our method is not limited to JT-VAE but can be applied with any latent-variable model. Furthermore, we use JT-VAE with all the baseline methods for a fair comparison.

We use Algorithm 1 with $\nu = 10$ and $B = 450$. In Fig. 4(b) we demonstrate that our method is able to identify candidate molecules that have a higher penalised logP value than competing methods. Furthermore, in Fig. 4(c), we demonstrate the performance of our method with different choices of auxiliary covariates for the additive kernel of the GP prior. The results show that it is indeed beneficial to include the additional auxiliary covariate information, whenever they are available in an application, along with the partially observed quantity of interest (penalised logP). See Suppl. Fig. 13 for a visualisation of the marginal variance of the additive components. We visualise the mean and standard deviation of the marginal variance for each of the kernel components across the 56 latent dimensions. In Suppl. Fig. 14, we demonstrate the performance of our method with fewer instances for which the penalised logP is observed. For this experiment, we assumed that the penalised $logP$ is

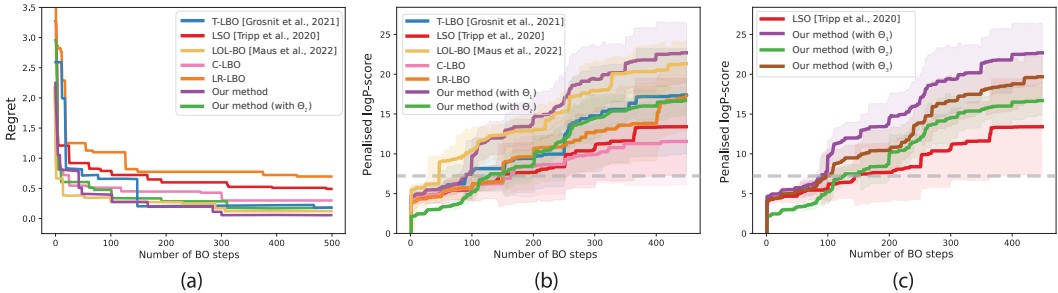

Figure 4: *Results from the expression reconstruction and molecule optimisation experiments.* (a) The mean regret achieved over 5 repetitions. $\Theta_2$ pertains to a kernel over only the target quantity. ***Lower values are better***. (b) The penalised logP score achieved by our method compared to competing methods. (c) Comparison of the penalised logP score achieved by our method using different choices of additional covariates in the additive kernel of the GP prior VAE. $\Theta_1$ pertains to an additive kernel over all 7 covariates and the partially observed target quantity, $\Theta_2$ to a kernel over only the partially observed target quantity, and $\Theta_3$ to an additive kernel over only the 5 additional properties that were calculated with RDKit (and does not include the partially observed target quantity). In figures (b) and (c), the mean penalised logP score over 10 repetitions is visualised together with the 95% confidence interval (shaded region). The grey line pertains to the highest penalised logP score in the training set. ***Higher values are better***

observed for only 0.1% of the data. The overall performance of all the methods decreases because the BO has fewer points to fit the surrogate model with. However, it is interesting that our method with an additive kernel that does not include the quantity of interest performs the best. We postulate that this is because only a few instances of the quantity of interest are observed and hence the estimated values for the unobserved quantities of interest have low quality. We believe that this demonstrates that our model can learn meaningful latent representations without making use of the partially observed quantity of interest and can be applied to datasets with only a few labelled instances.

In Suppl. Fig 15, we visualise the latent space using t-SNE (Van der Maaten & Hinton, 2008) and colour the latent embedding by the respective molecular properties. We note that the model learns a latent embedding that changes smoothly with respect to the target quantity as well as with the respect to the additional covariates.

## 5.4 Evaluating Latent Space Structure for Gaussian Processes

We evaluate our model's ability to construct meaningful discriminative latent spaces for Gaussian processes (GPs). Following an approach similar to Grosnit et al. (2021), we leverage the trained encoder to map data points from the original space, $\boldsymbol{y} \in \mathcal{Y}$, onto a low-dimensional latent space, $\boldsymbol{z} \in \mathcal{Z} = \mathbb{R}^L$, where we fit a GP using the original labels. To assess whether structured latent representations enhance GP generalisation, we ensure a unified experimental setup across all tasks. Specifically, we use 80% of the encoded latent points (from the respective training splits) to train a sparse GP with 500 inducing points and compute the predictive log-likelihood on the remaining 20% of held-out data. This experiment provides insights into the impact of clustered latent inputs on GP regression—an essential factor in the Bayesian Optimisation (BO) process. In Suppl. Table 2, we show that our method achieves the highest predictive log-likelihood, highlighting how the discriminative latent space enhances GP generalisation.

## 6 Conclusion

In this paper, we proposed a novel GP prior VAE-based method to perform high-dimensional BO. We demonstrated the efficacy of our method on simulated datasets as well as in the discovery of novel molecules that optimise a quantity of interest. Our method shows that it can be beneficial to include auxiliary covariates (even partially observed) for performing BO in the latent space. Furthermore, our approach can efficiently handle partially observed target quantities. Given the flexibility and performance of our model, we expect our approach to be beneficial to scalable BO.

ACKNOWLEDGMENTS

We gratefully acknowledge the computational resources provided by Aalto Science-IT, Finland. Our sincere thanks also go to Charles Gadd for his valuable discussions.

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

# Appendices

## A  DERIVATION OF THE ELBO

Following (Kingma & Welling, 2014), the ELBO of the marginal log-likelihood for the standard VAE model can be written as:

$$\log p_\omega(Y) \geq \mathcal{L}(\phi, \psi, \theta; Y) \triangleq \sum_{n=1}^{N} \mathbb{E}_{q_\phi}[\log p_\psi(\boldsymbol{y}_n|\boldsymbol{z}_n)] - \mathrm{KL}[q_\phi(\boldsymbol{z}_n|\boldsymbol{y}_n)||p_\theta(\boldsymbol{z}_n)] \to \max_{\phi, \psi, \theta}, \quad (2)$$

where $\phi$ and $\psi$ are the encoder and decoder weights respectively, $\boldsymbol{z}$ refers to the low-dimensional latent representation, $\boldsymbol{y}$ refers to the observations, and KL refers to the Kullback-Leibler divergence.

Since we use GP prior VAEs that assume independent priors for each of the latent dimensions, we write the joint conditional prior as:

$$p_\theta(Z \mid X) = \prod_{l=1}^{L} p_\theta(\bar{\boldsymbol{z}}_l \mid X) = \prod_{l=1}^{L} \mathcal{N}\left(\bar{\boldsymbol{z}}_l \mid \boldsymbol{0}, K_{XX}^{(l)}\right),$$

where $\bar{\boldsymbol{z}}_l = \tau_l(X) = (\tau_l(\boldsymbol{x}_1), \ldots, \tau_l(\boldsymbol{x}_N))^T$, $\tau_l(\boldsymbol{x}) \sim \mathcal{GP}(\mu_l(\boldsymbol{x}), k_l(\boldsymbol{x}, \boldsymbol{x}' \mid \theta_l))$ such that $\mu_l(\boldsymbol{x})$ is the mean, $k_l(\boldsymbol{x}, \boldsymbol{x}' \mid \theta_l)$ is the covariance function and $\theta_l$ denotes the parameters of the covariance function, and $K_{XX}^{(l)}$ is a $N \times N$ covariance matrix for the $l^{\text{th}}$ latent dimension.

We summarise the observed and unobserved parts as $Y = (Y^\text{o}, Y^\text{u})$ and $\tilde{X} = (\tilde{X}^\text{o}, \tilde{X}^\text{u})$ and write the ELBO as:

$$\log p_\gamma(Y^\text{o}|\tilde{X}^\text{o}) \geq \underbrace{\mathbb{E}_q[\log p_\psi(Y^\text{o}|Z)] - \mathrm{KL}[q_\phi(Z, \tilde{X}^\text{u}|Y^\text{o}, \tilde{X}^\text{o})||p_{\theta,\lambda}(Z, \tilde{X}^\text{u}|\tilde{X}^\text{o})]}_{\triangleq \mathcal{L}(\phi, \psi, \theta, \lambda; Y^\text{o}, \tilde{X}^\text{o})}. \quad (3)$$

We use a conditionally independent factorisable variational approximation:

$$q_\phi(Z, \tilde{X}^\text{u}|Y^\text{o}, \tilde{X}^\text{o}) = q_\phi(Z|Y^\text{o}, \tilde{X}^\text{o})q_\phi(\tilde{X}^\text{u}|\tilde{X}^\text{o}) = \prod_{i=1}^{N} q_\phi(\boldsymbol{z}_i|\boldsymbol{y}_i^\text{o}, \tilde{\boldsymbol{x}}_i^\text{o})q_\phi(\tilde{\boldsymbol{x}}_i^\text{u}|\tilde{\boldsymbol{x}}_i^\text{o}). \quad (4)$$

We assume that $q_\phi(\boldsymbol{z}_i|\boldsymbol{y}_i^\text{o}, \tilde{\boldsymbol{x}}_i^\text{o})$ factorises also across the latent dimensions, which allows us to write the variational approximation alternatively as

$$q_\phi(Z, \tilde{X}^\text{u}|Y^\text{o}, \tilde{X}^\text{o}) = q_\phi(Z|Y^\text{o}, \tilde{X}^\text{o})q_\phi(\tilde{X}^\text{u}|\tilde{X}^\text{o}) = \prod_{l=1}^{L} q_\phi(\bar{\boldsymbol{z}}_l|Y^\text{o}, \tilde{X}^\text{o}) \prod_{i=1}^{N} q_\phi(\tilde{\boldsymbol{x}}_i^\text{u}|\tilde{\boldsymbol{x}}_i^\text{o}). \quad (5)$$

Following Ramchandran et al. (2024), we simplify the KL term in Eq. 3 as:

$$\mathrm{KL}[q_\phi(Z, \tilde{X}^\text{u}|Y^\text{o}, \tilde{X}^\text{o})||p_{\theta,\lambda}(Z, \tilde{X}^\text{u}|\tilde{X}^\text{o})]$$

$$= \mathbb{E}_{q_\phi}\left[\mathrm{KL}[q_\phi(Z|Y^\text{o}, \tilde{X}^\text{o})||p_\theta(Z|\tilde{X}^\text{u}, \tilde{X}^\text{o})]\right] + \mathrm{KL}[q_\phi(\tilde{X}^\text{u}|\tilde{X}^\text{o})||p_\lambda(\tilde{X}^\text{u}|\tilde{X}^\text{o})]$$

$$= \sum_{l=1}^{L} \underbrace{\mathbb{E}_{q_\phi}\left[\mathrm{KL}[q_\phi(\bar{\boldsymbol{z}}_l|Y^\text{o}, \tilde{X}^\text{o})||p_\theta(\bar{\boldsymbol{z}}_l|\tilde{X}^\text{u}, \tilde{X}^\text{o})]\right]}_{\leq D_{\mathrm{KL}}^1} + \sum_{i=1}^{N} \underbrace{\mathrm{KL}[q_\phi(\tilde{\boldsymbol{x}}_i^\text{u}|\tilde{\boldsymbol{x}}_i^\text{o})||p_\lambda(\tilde{\boldsymbol{x}}_i^\text{u}|\tilde{\boldsymbol{x}}_i^\text{o})]}_{D_{\mathrm{KL}}^2} \quad (6)$$

by using the assumption of a factorising latent space, $p_\theta(Z|\tilde{X}^\text{u}, \tilde{X}^\text{o}) = \prod_{l=1}^{L} p_\theta(\bar{\boldsymbol{z}}_l|\tilde{X}^\text{u}, \tilde{X}^\text{o})$ and a mean-field normal posterior for $q_\phi(\bar{\boldsymbol{z}}_l|Y^o, \tilde{X}^\text{o})$, with a variational mean $\bar{\boldsymbol{\mu}}_l = (\mu_{\phi,l}(\tilde{\boldsymbol{x}}_1^\text{o}, \boldsymbol{y}_1^\text{o}), \ldots, \mu_{\phi,l}(\tilde{\boldsymbol{x}}_N^\text{o}, \boldsymbol{y}_N^\text{o}))^T$ and a covariance matrix $W_l = \mathrm{diag}(\sigma_{\phi,l}^2(\tilde{\boldsymbol{x}}_1^\text{o}, \boldsymbol{y}_1^\text{o}), \ldots, \sigma_{\phi,l}^2(\tilde{\boldsymbol{x}}_N^\text{o}, \boldsymbol{y}_N^\text{o}))$ for the $l^{\text{th}}$ latent dimension. The expectation in Eq. 6 is w.r.t. the unobserved auxiliary covariates $\tilde{X}^\text{u}$ that are the inputs to the GP kernel and, therefore, the expectation does not have a closed form but can be approximated by Monte Carlo sampling.

There are several different approaches to approximate the GP prior in order for the ELBO in Eq. 6 to scale to large datasets. We make use of a mini-batch compatible approach proposed by Ramchandran et al. (2021) that uses the inducing point method (Titsias, 2009; Hensman et al., 2013) and exploits the structure of the GP prior, as described in the next section.

## A.1 SCALABLE COMPUTATION AND MINIBATCHING

Each of the KL divergences $\text{KL}[q_\phi(\bar{z}_l|Y^o, \tilde{X}^o)||p_\theta(\bar{z}_l|\tilde{X}^u, \tilde{X}^o)]$ in Eq. 6 has a computation complexity of $\mathcal{O}(N^3)$. Below we drop the index of the latent dimension, $l$, for simplicity. Relying on the derivation proposed by Ramchandran et al. (2021) to obtain a scalable ELBO, we use the low-rank inducing point approximation for GPs and use $M$ inducing locations $S = (s_1, \ldots, s_M)$ in $\mathcal{X}$ and the corresponding inducing function values $u_l = (\tau_l(s_1), \ldots, \tau_l(s_M))^T = (u_{l1}, \ldots, u_{lM})^T$ for each latent dimension (Hensman et al., 2013). We explicitly keep track of the distribution of the Gaussian inducing values $u_l \sim \mathcal{N}(m_l, H_l)$, where $m_l$ and $H_l$ are global variational parameters. We can then derive an upper-bound for the KL divergence $\text{KL}[\mathcal{N}(\bar{\mu}, W)||\mathcal{N}(0, K_{\tilde{X}\tilde{X}})] \leq D_{\text{KL}}^1$ as well as an unbiased, batch-normalised partial sum over a subset of indices, $\mathcal{I} \subset \{1, \ldots, N\}$ of size $|\mathcal{I}| = \hat{N}$ such that $\hat{D}_{\text{KL}}^1 \approx D_{\text{KL}}^1$, where

$$
\begin{aligned}
\hat{D}_{\text{KL}}^1 = \frac{1}{2}\frac{N}{\hat{N}} \sum_{i \in \mathcal{I}} &\Big( \sigma_z^{-2}(K_{\tilde{x}_i S} K_{SS}^{-1} m - \bar{\mu}_i)^2 + \sigma_z^{-2}\sigma_i^2 + \sigma_z^{-2}\tilde{K}_{ii} \\
&+ \sigma_z^{-2}\text{tr}\left( (K_{SS}^{-1} H K_{SS}^{-1})(K_{S\tilde{x}_i} K_{\tilde{x}_i S}) \right) - \log \sigma_i^2 \Big) + \frac{N}{2}\log \sigma_z^2 - \frac{N}{2} \\
&+ \text{KL}[\mathcal{N}(m, H)||\mathcal{N}(0, K_{SS})],
\end{aligned}
\tag{7}
$$

where $\bar{\mu}_i = \bar{\mu}_\phi(\tilde{x}_i, y_i^o)$ and $\sigma_i^2 = \sigma_\phi^2(\tilde{x}_i, y_i^o)$ are the encoder means and variances, $\tilde{K}_{ii}$ denotes the $i^{\text{th}}$ diagonal element of $\tilde{K} = K_{\tilde{X}\tilde{X}} - K_{\tilde{X}S} K_{SS}^{-1} K_{S\tilde{X}}$, and $K_{SS}$ as well as $K_{\tilde{x}_i S} = K_{S\tilde{x}_i}^T$ are defined similarly as $K_{\tilde{X}\tilde{X}}$. The conditional probability $p_\lambda(\tilde{X}^u|\tilde{X}^o)$ in Eq. 6 simplifies to $p_\lambda(\tilde{X}^u|\tilde{X}^o) = p_\lambda(\tilde{X}^u)$. As described in (Ramchandran et al., 2024), $p_\lambda(\tilde{X}^u)$ can be an informative prior and $D_{\text{KL}}^2$ is amenable to mini-batching.

Therefore, the ELBO for GP prior VAE models that marginalises missing covariates and affords efficient optimisation with stochastic gradient descent is obtained from the Eqs. 3, 6, 7. For a more detailed derivation, please refer to Ramchandran et al. (2021; 2024).

## B THE EXPECTED IMPROVEMENT ACQUISITION FUNCTION

The expected improvement acquisition function estimates improvement that would be achieved when choosing a specific point $z$ as the next point to query. It balances between exploration and exploitation of the black-box function $f$. The expected improvement (EI) is defined as

$$
\alpha(z) = \text{EI}(z) = \int_{-\infty}^{\infty} \underbrace{\max(f(z) - f^*), 0)}_{\text{Improvement}} \varphi\left( \frac{f(z) - \mu(z)}{\sigma(z)} \right) df(z),
$$

where $f^*$ is our current optimum, $y$ is an instance in the data space, $\hat{y} \leftarrow g_{\psi^*}(\cdot|\hat{z})$ where $\psi^*$ pertains to the trained decoder weights and $\hat{z}$ is the chosen latent space location, and $\varphi(t)$ is the probability density function of the standard normal distribution, $\mathcal{N}(0, 1)$. The expected improvement can be analytically evaluated under the GP surrogate model:

$$
\text{EI}(z) = \begin{cases} \underbrace{(\mu(z) - f(\hat{y}) - \xi)\Phi(T)}_{(i)} + \underbrace{\sigma(z)\varphi(T)}_{(ii)} & \text{if } \sigma(z) > 0 \\ 0 & \text{if } \sigma(z) = 0 \end{cases}
\tag{8}
$$

where,

$$
T = \begin{cases} \frac{\mu(z) - f(\hat{y}) - \xi}{\sigma(z)} & \text{if } \sigma(z) > 0 \\ 0 & \text{if } \sigma(z) = 0. \end{cases}
$$

In the above equation, $\mu(z)$ and $\sigma(z)$ are the mean and standard deviation of the surrogate GP posterior predictive at $z$. $\Phi$ and $\varphi$ are the cumulative distributive function and probability density function of the standard normal distribution, respectively.

In Eq. 8, part $(i)$ corresponds to the exploitation term and part $(ii)$ corresponds to the exploration term. The parameter $\xi$ controls the amount of trade-off between exploration and exploitation (higher values $\xi$ leads to more exploration). We set $\xi$ to be 0.01 in all our experiments (a recommended default value). For a detailed review of the expected improvement and other popular acquisition functions we refer the reader to (Frazier, 2018; Garnett, 2023).

## C  OPTIMISATION AND PRACTICAL CONSIDERATIONS

To maximise the evidence lower bound, we use the Adam optimiser (Kingma & Ba, 2015), which is an adaptive learning rate method that maintains an exponentially decaying average of past gradients as well as squared gradients. The parameters that need to be optimised include the neural network weights for the encoder ($\phi$) as well as decoder ($\psi$) and the GP kernel parameters ($\theta$). Moreover, we separately fit the GP surrogate model for the Bayesian optimisation. In the case of mini-batch training, the optimisation steps are conducted interchangeably with natural gradient-based updates of the variational parameters.

We use PyTorch (Paszke et al., 2019) for the inference implementation which allows the computation of derivatives using automatic differentiation and we use BoTorch (Balandat et al., 2020) for Bayesian optimisation. For all experiments we set the frequency of retraining $\nu = 10$ and the stopping criterion $\eta = 0.1$. We set the number of latent dimensions to 8 for the synthetic dataset experiment, 25 for the expression reconstruction experiment, and 56 for the molecular discovery experiment.

## D  EXPERIMENT WITH VANILLA VAE BO

We demonstrate the performance of Vanilla VAE BO (i.e. no weighted retraining) using synthetic data. From Suppl. Fig. 5, we can clearly see that our method as well as the other baselines demonstrate better performance.

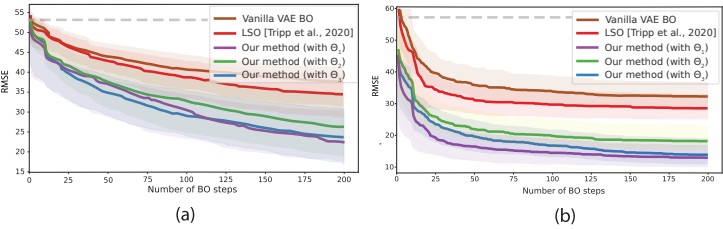

Figure 5: *Results from our experiments with a synthetic dataset. **Lower values are better**.* (a) Comparing the performance of our model with the LSO benchmark. The dataset comprises 500 instances out of which the target quantity is observed only for 100 instances. $\Theta_1$ pertains to an additive GP prior VAE over all three covariates $x$ and the partially observed quantity of interest $c$, $\Theta_2$ to a GP prior VAE over only the partially observed target $c$, and $\Theta_3$ to an additive GP prior VAE over the partially observed quantity of interest $c$ and shift$_x$. (b) Similarly, we also demonstrate our model's performance on a dataset which comprises 5000 instances out of which the quantity of interest is observed only for 500 instances. All plots depict the mean quantity of interest value with the 95% confidence interval (shaded region) obtained over 100 repetitions with regenerated training data and target images. The grey line pertains to the lowest RMSE in the training set.

## E  ABLATION STUDY

We demonstrate how our method performs with different subsets of additional auxiliary covariates. The ablations were run on the simulated data with 5000 samples and the target quantity of interest was observed for 500 samples. Fig. 6 depicts the mean target quantity of interest obtained over 100 repetitions with regenerated training data and target images.

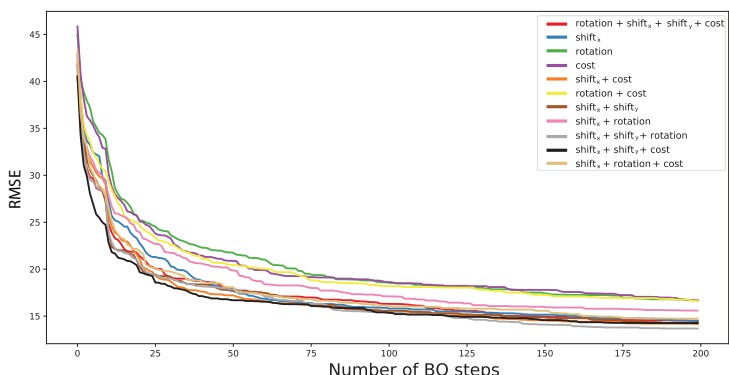

Figure 6: *Results from experimenting with the choice of kernel for the synthetic dataset.*

# F    NEURAL NETWORK ARCHITECTURES

## F.1    SYNTHETIC DATASET

Table 1: Neural network architectures used in the simulated dataset.

| | Hyperparameter | Value |
|---|---|---|
| | Dimensionality of input | $52 \times 52$ |
| | Number of convolution layers | 3 |
| | Number of filters per convolution layer | 144 |
| | Kernel size | $3 \times 3$ |
| | Stride | 2 |
| Inference | Pooling | Max pooling |
| network | Pooling kernel size | $2 \times 2$ |
| | Pooling stride | 2 |
| | Number of feedforward layers | 2 |
| | Width of feedforward layers | 500, 50 |
| | Dimensionality of latent space | 8 |
| | Activation function of layers | RELU |
| | Dimensionality of input | 8 |
| | Number of transposed convolution layers | 3 |
| | Number of filters per transposed convolution layer | 256 |
| Generative | Kernel size | $4 \times 4$ |
| network | Stride | 2 |
| | Padding | 2 |
| | Number of feedforward layers | 2 |
| | Width of feedforward layers | 50, 500 |
| | Activation function of layers | RELU |

In the synthetic data experiment, we use the neural network architecture described in Table 1.

## F.2    EXPRESSION RECONSTRUCTION

In the expression reconstruction experiment we use the Grammar VAE (Kusner et al., 2017) which is a computational model used in natural language processing and generative modelling. It combines principles from VAEs and context-free grammars to learn and generate structured sequences of symbols, such as sentences, mathematical expressions, or code.

In other words, the Grammar VAE extends the VAE framework to handle structured data where the order and relationships between elements matter. Furthermore, Grammar VAE incorporates context-free grammars which define the syntax and structure of sequences. This allows the model to

capture the hierarchical and compositional nature of sequences, making it well-suited for generating structured outputs. In addition to this, Kusner et al. (2017) proposed to represent the discrete data using a parse tree from the context-free grammar. Therefore, the model is a variational autoencoder which encodes and decodes directly to and from the generated parse trees while ensuring that the generated outputs are always valid.

In our experiments, we use a latent space dimension of 25 and make use of the neural network architecture specified in (Kusner et al., 2017).

### F.3 MOLECULE OPTIMISATION

In the molecule generation experiment we use the Junction Tree VAE (JT-VAE) (Jin et al., 2018) which extends VAEs to molecular graphs by introducing a suitable encoder and decoder. The encoder learns two latent representations: one that encodes the tree structure and high-level cluster information while the other encodes fine-grained connectivity details. In particular, the model generates a molecular graph in two phases: first it generates a tree-structured scaffold over chemical sub-structures and then combines them into molecules with a graph message passing network. The molecule is encoded into two latent representations: $z = [z_\mathcal{T}, z_G]$ where $z_\mathcal{T}$ encodes the tree structure and the information of the clusters that are in the tree without fully capturing how exactly the clusters are mutually connected. The graph to capture the fine-grained connectivity is encoded by $z_G$.

The latent representation is then decoded back into a molecular graph in two stages. First, reproduce the junction tree using a tree decoder. Then, predict the fine-grained connectivity between the clusters in the junction tree using a graph decoder to obtain the full molecular graph. The decoder generates the molecule piece-by-piece utilising the components and how they interact instead of assembling the molecule atom-by-atom and/or through chemically invalid intermediaries.

The graph encoder is a graph message passing network (graph neural networks), the tree encoder is a tree message passing network (related to RNNs and tree-LSTM), the junction tree is reconstructed using a structured tree decoder, and the fine-grained cluster details are obtained using a graph decoder. Furthermore, the latent space dimension is 56 (tree and graph representations are 28 dimensions each).

The key benefit is the incremental expansion of the molecule while maintaining chemical validity at every step. Furthermore, each molecule is built from sub-graphs chosen out of a vocabulary of valid components. In this work, we use the same neural network architecture specification as in (Jin et al., 2018). Our proposed model is agnostic to the choice of the underlying neural network architecture.

## G SUPPLEMENTARY IMAGES

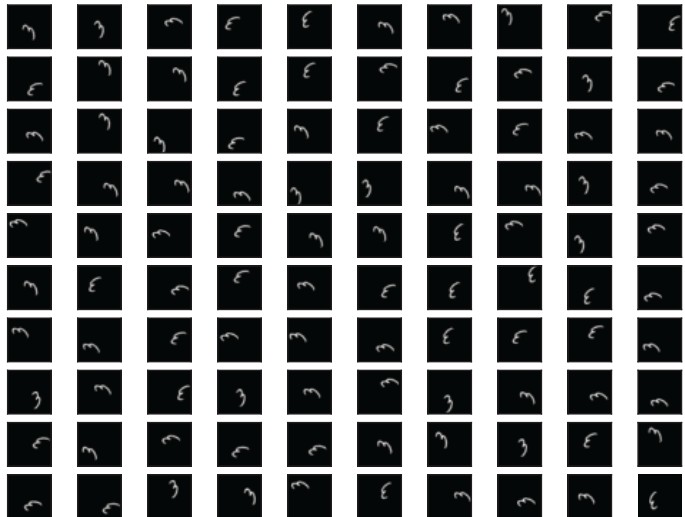

Figure 7: *A random sample of digits (from the the synthetic dataset) that have been rotated and shifted along the x and y axis.*

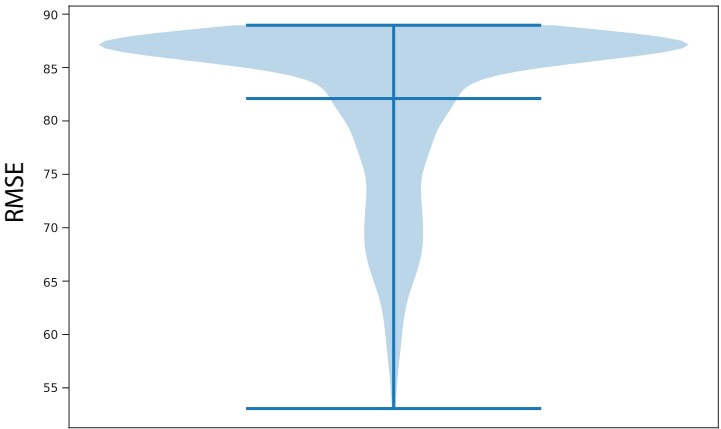

Figure 8: *Violin plot visualising the distribution of the quantity of interest in an instance of the synthetic dataset.*

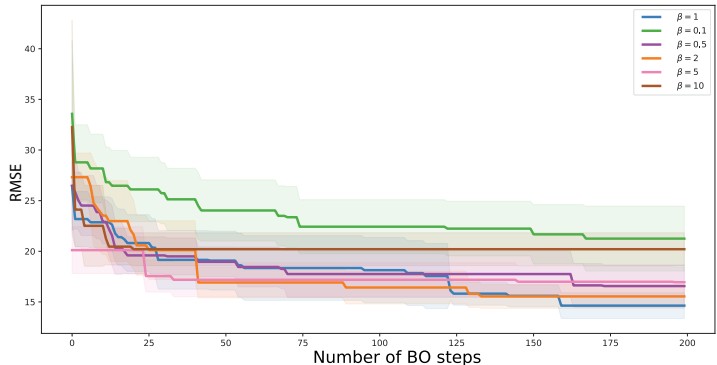

Figure 9: *A demonstration of the effect of $\beta$ (as in $\beta$-VAE (Higgins et al., 2017)) on the model performance in the synthetic dataset.*

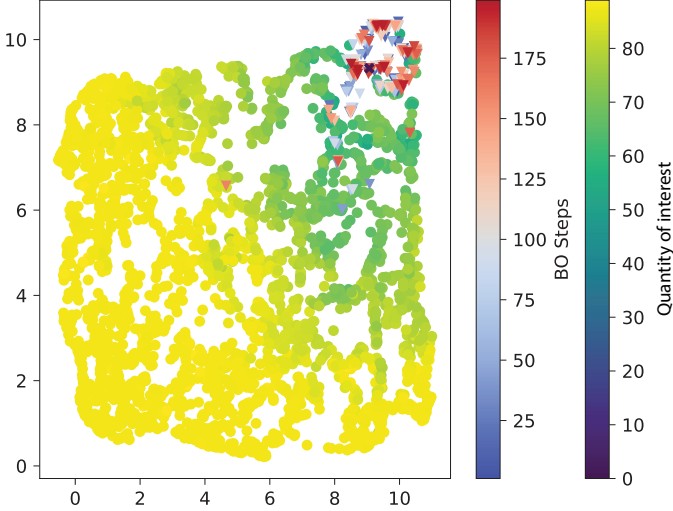

Figure 10: *Visualisation of the latent space in the synthetic data experiment.* We performed a projection of the latent space down to two dimensions using UMAP (McInnes et al., 2018). The inverted triangles refer to the BO steps and the blue cross refers to the latent space representation of the "optimal" instance which we hope our method would find (not included in the training set). The latent embedding is coloured by the quantity of interest.

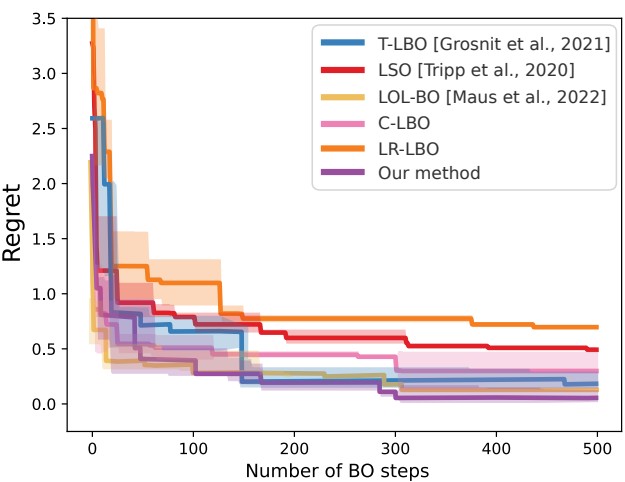

Figure 11: *Results from the expression reconstruction experiment.* The mean regret achieved by our method compared to competing methods over 5 repetitions is visualised together with the 95% confidence interval (shaded region). ***Lower values are better***.

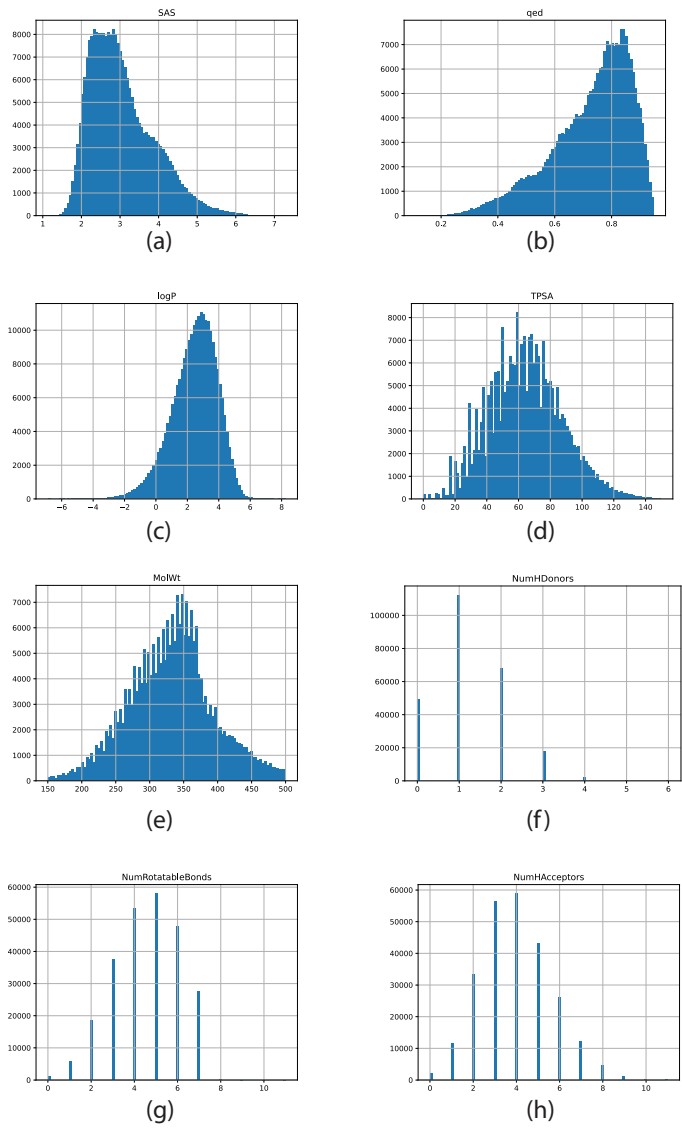

Figure 12: *Histograms visualising the distribution of the properties in the ZINC-250K dataset.*

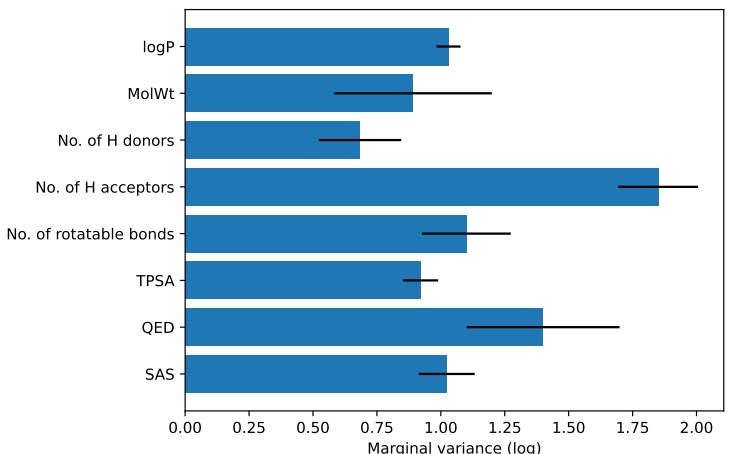

Figure 13: *Marginal variance of the additive components in the molecule discovery experiment.* We visualise the mean and standard deviation of the marginal variance for each of the kernel components across the 56 latent dimensions. This pertains to an additive kernel over all the covariates.

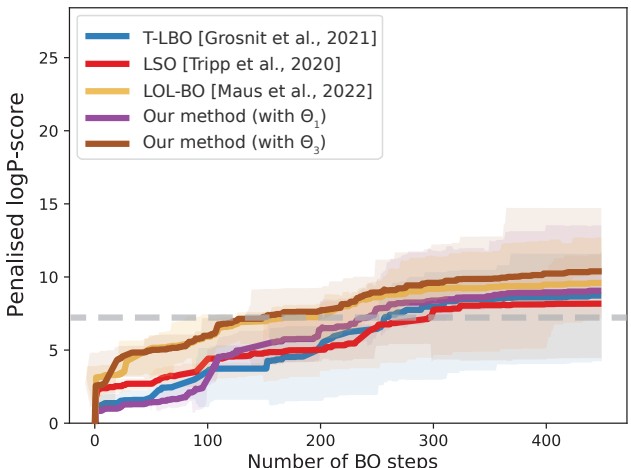

Figure 14: *Results from the molecule optimisation experiment with the penalised logP observed for only* $0.1\%$ *of the data.* The mean penalised logP score over 10 repetitions is visualised together with the $95\%$ confidence interval (shaded region). The grey line pertains to the highest penalised logP score in the training set. $\Theta_1$ pertains to an additive kernel over all 7 covariates and the partially observed target quantity, and $\Theta_3$ pertains to an additive kernel over only the 5 additional properties that were calculated with RDKit (and does not include the partially observed target quantity). The overall performance of all the methods decrease because the Bayesian optimisation has fewer number of points to fit the surrogate model with. However, it is interesting to note that our method with an additive kernel that does not include the quantity of interest performs the best (by a small margin). The other approaches make use of the quantity of interest (penalised logP) while fitting the model. We believe that this demonstrates that our model (with $\Theta_3$) is able to learn meaningful latent representations without making use of the partially observed target quantity of interest. ***Higher values are better***.

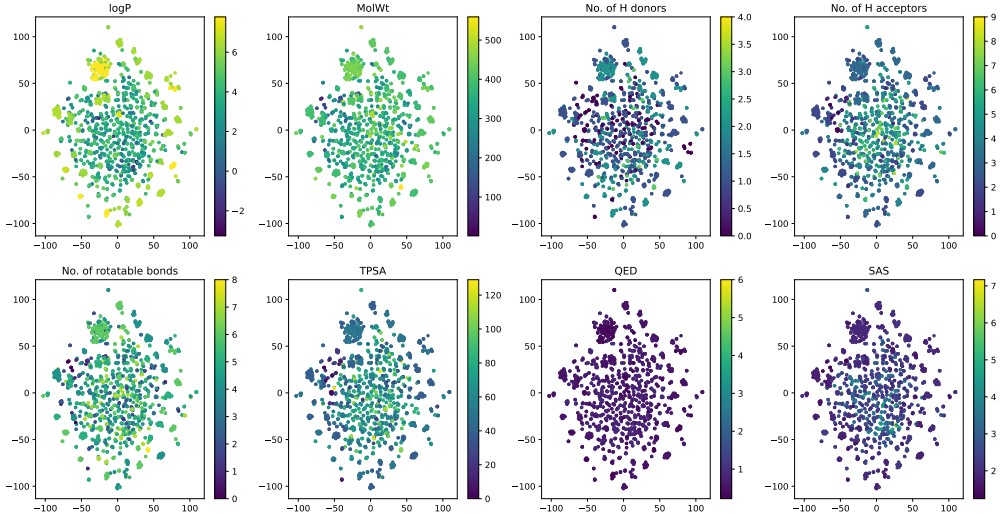

Figure 15: *Visualisation of the latent space in the molecule discovery experiment.* We performed a projection of the latent space from 56 dimensions down to two dimensions using t-SNE (Van der Maaten & Hinton, 2008) and using the validation dataset for convenience. The latent embedding is coloured by the respective molecular properties. The proposed model learns a latent embedding that changes smoothly with respect to the target quantity as well as with respect to the additional covariates (noting again that the visualisation corresponds to a 2-D t-SNE embedding).

## H    Supplementary Tables

Table 2: This table illustrates how separation in the latent space enhances GP generalisation. It reports the GP predictive log-likelihood on the held-out validation sets, along with the standard deviation ($\pm$) on the validation set. Higher (less negative) values are better.

|  | Expression reconstruction | Molecular discovery |
|---|---|---|
| LSO (Tripp et al., 2020) | $-3.1 \pm 0.09$ | $-2.01 \pm 0.25$ |
| T-LBO (Grosnit et al., 2021) | $-1.85 \pm 0.07$ | $-1.49 \pm 0.29$ |
| LOL-BO (Maus et al., 2022) | $-1.75 \pm 0.07$ | $-1.39 \pm 0.25$ |
| Our method | $\mathbf{-1.72 \pm 0.08}$ | $\mathbf{-1.37 \pm 0.27}$ |

Table 3: Average run time / wall clock time. In the synthetic dataset experiment $200$ BO steps are performed and in the molecule discovery experiment $450$ BO steps are performed.

| Method | Experiment | Configuration | GPU type | CPU type | Runtime (avg.) |
|---|---|---|---|---|---|
| Our method | Synthetic data (5000 obs.) | Kernel $\Theta_1$ | AMD MI250x | AMD EPYC "Trento" | 152 mins |
|  |  | Kernel $\Theta_2$ | AMD MI250x | AMD EPYC "Trento" | 140 mins |
|  |  | Kernel $\Theta_3$ | AMD MI250x | AMD EPYC "Trento" | 163 mins |
|  | Expression reconstruction | - | Nvidia Tesla V100 | Intel Xeon Gold 6134 | 1064 mins |
|  | Molecular discovery | Kernel $\Theta_1$ | Nvidia Tesla V100 | Intel Xeon Gold 6134 | 1682 mins |
|  |  | Kernel $\Theta_2$ | Nvidia Tesla V100 | Intel Xeon Gold 6134 | 1641 mins |
|  |  | Kernel $\Theta_3$ | Nvidia Tesla V100 | Intel Xeon Gold 6134 | 1668 mins |
| LSO (Tripp et al., 2020) | Synthetic data (5000 obs.) | - | AMD MI250x | AMD EPYC "Trento" | 102 mins |
|  | Expression reconstruction | - | Nvidia Tesla V100 | Intel Xeon Gold 6134 | 723 mins |
|  | Molecular discovery | - | Nvidia Tesla V100 | Intel Xeon Gold 6134 | 1038 mins |
| T-LBO (Grosnit et al., 2021) | Expression reconstruction | - | Nvidia Tesla V100 | Intel Xeon Gold 6134 | 918 mins |
|  | Molecular discovery | - | Nvidia Tesla V100 | Intel Xeon Gold 6134 | 1582 mins |
| LOL-BO (Maus et al., 2022) | Expression reconstruction | - | Nvidia Tesla V100 | Intel Xeon Gold 6134 | 802 mins |
|  | Molecular discovery | - | Nvidia Tesla V100 | Intel Xeon Gold 6134 | 845 mins |

## I    Limitations

While our method proposes a novel approach to performing high-dimensional Bayesian optimisation efficiently, it shares several of the limitations of standard VAEs. For example:

- It can be challenging to model complex (or multi-modal) data.
- The performance is dependent on the expressiveness of the chosen neural network architecture for the encoder and decoder.
- The latent space is assumed to follow a Gaussian distribution which may not hold true for all datasets.
- Sensitivity to the hyperparameter values such as dimensionality of the latent space, weight of the KL divergence ($\beta$), minibatch size, etc.

Furthermore, in GP prior VAEs, the choice of the auxiliary covariates used for the GP prior needs to be done empirically. Despite these limitations, GP prior VAEs have been successful in various applications and have contributed to advances in generative modelling and unsupervised representation learning.

## J    Broader Impacts

Generative machine learning models have gained significant attention in recent times. In this work, we make use of the variational autoencoder which has been primarily used for representation learning, imputation, and data generation tasks. However, VAEs (and deep generative models in general) present several societal implications that extend beyond the scope of academia and research. Furthermore, they present ethical considerations due to their potential malicious applications including contributing

to misinformation and possible privacy concerns. Robust frameworks as well as guidelines need to be established to address these concerns and to ensure the responsible deployment of generative machine learning technologies. It is essential to navigate the ethical concerns and to ensure the responsible use of deep generative models for the betterment of society.

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
