# OpenReview forum: "High-Dimensional Bayesian Optimisation with Gaussian Process Prior Variational Autoencoders"
_ICLR.cc/2025/Conference — ICLR 2025 Poster_

### Official Review · Reviewer_t5E1 · 2024-10-18

**Soundness:** 3
**Presentation:** 3
**Contribution:** 2
**Rating:** 5
**Confidence:** 3

**Summary:**

This paper introduces a novel approach to Bayesian optimization (BO) that addresses high-dimensional data incorporating auxiliary covariates, even when some of the covariates contain missing values.  The method builds upon a variational autoencoder (VAE) where the latents follow a Gaussian Process (GP) prior. The approach is technically sound, and the experimental results show its effectiveness in both synthetic and real-world settings.

However, the paper's novelty is somewhat limited, as it builds on existing GP prior VAE models with missing covariates. Furthermore, the absence of a key baseline in one of the experiments raises concerns about the completeness of the experimental evaluation. Addressing these issues—especially by adding the missing baseline—would make a stronger case for the proposed method.

If the authors include this baseline in the first experiment and their method shows significant improvements (or they provide a justified reason for its absence), I would be inclined to increase my evaluation score from a weak reject to a weak accept.

**Strengths:**

**1. Clarity and Readability:** The paper is well-structured and clearly written. The authors explain complex ideas in an accessible way, ensuring that the technical details are easy to follow. I am not an expert for BO but could get a good understanding of the paper within a few hours.

**2. Technical Rigor:** The paper is technically solid. The authors carefully describe the underlaying probabilistic model step-by-step. The integration of auxiliary covariates, even when some values are missing, is handling in a principled manner by applying variational inference.

**3. Experimental Thoroughness:** The experiments are in general carefully executed. This was the part of the paper that I enjoyed reading the most. They span multiple datasets from different domains and the results are studied in-depth.

**Weaknesses:**

**1. Limited Novelty:** While the application of the GP prior VAE to Bayesian optimization is novel, the underlying model itself—GP prior VAE with partial observations —has been published previously in Ramchandran et al. (2024). This implies that the core contribution of the paper lies primarily in applying the method to the BO context. This limits the overall novelty of the work.

**2. Competitive/Missing Baselines:** The method LOL-BO (Maus et al., 2022) shows competitive performance on the expression reconstruction dataset, and is only marginally outperformed on the molecular optimization experiment. However, in the first experiment on synthetic data, I found this method missing. It is important for the authors to include this baseline in this experiment to ensure a fair comparison. If their approach significantly outperforms this baseline, it would strengthen the contribution. If there is a specific reason why this approach cannot be applied for this experiments, it needs to be stated more clearly in the paper.

**3. Scalability Dependency:** The scalability of the method relies on leveraging sparse Gaussian Process (GP) techniques, using inducing points to approximate the GP. However, selecting meaningful inducing points becomes challenging when the dataset changes over time, as it is the case in BO applications. As new data arrives, it is unclear how to update the inducing points in a principled and computationally efficient manner. This is a critical issue for ensuring that the model scales well over BO iterations, and the paper should at least discuss this problem.

**Questions:**

See weaknesses.

---

> ### Author Response · Authors · 2024-11-24
>
> **"Limited novelty"**
>
> Our approach innovatively applies the principles of Bayesian optimisation (BO) within a low-dimensional latent space, introducing a nuanced method that consistently enhances efficiency and performance. Central to our contribution is the use of conditional VAE-based methods that makes use of all available information (i.e. auxiliary covariates and target quantity values that may or may not be fully observed) to generate highly informative latent representations. These representations, in turn, facilitate more effective BO task outcomes. To the best of our knowledge, we are the first who propose to utilise Gaussian process (GP) prior VAEs, or in fact any conditional VAEs, for high-dimensional BO.
>
> We position our method as a sophisticated alternative to traditional metric loss approaches. This distinction is underscored by the flexibility of our model, as detailed in Algorithm 1 of our manuscript. Specifically, our model demonstrates a remarkable adaptability to a variety of BO components, underlining its versatility and broad applicability.
>
> **"Competitive/missing baselines"**
>
> The synthetic experiment described in Section 5.1 was designed to illustrate the functioning of our proposed method and should be viewed as an illustrative toy example rather than a comparative study. The non-standard nature of this experimental setup was not easily adaptable for the competing methods, which is why they were not included in this part of the evaluation. Specifically, the use of a convolution-based neural network architecture (detailed in Table 1 of the Appendices) posed challenges for employing LOL-BO, as it was not designed for image-based data. That said, we have demonstrated the effectiveness of our method in comparison to others through more complex and widely recognised benchmark tasks that are common in the literature.
>
> Further development of existing methods falls outside the scope of our work, and we have employed the baseline methods as implemented in their published reference versions. The primary goal of our study was to propose an alternative approach that enhances existing VAE BO methods by leveraging the advantages of GP prior VAEs. While we acknowledge that the baseline methods also perform competitively, we believe the novelty of our work lies in the innovative integration of Gaussian Process priors.
>
> **"Scalability dependency"**
>
> We thank the reviewer for their insightful comment. To provide further context for the proposed method, BO is typically employed in scenarios where collecting observations is costly, time-consuming, or both. As a result, BO iterations are generally slow, and runtime considerations are less critical. Importantly, the number of new samples generated during BO is usually a small fraction of the overall training set size.
>
> As outlined in Algorithm 1 of our manuscript, we re-optimise the GP prior VAE model with the augmented dataset, incorporating the newly acquired samples after every $\nu$ BO steps. Consequently, the inducing points are also re-optimised, as described in Eq. 7 in Section A.1 of the Appendices.
>
> If observations are expensive to collect but can be obtained quickly, runtime efficiency becomes crucial. In such cases, gradients can be used to simultaneously optimise the inducing points while fitting the encoder and decoder parameters. Alternatively, an efficient approach is to take a random Monte Carlo sample of the available input data points at each BO iteration.

---

> > ### Comment · Reviewer_t5E1 · 2024-11-26
> >
> > Thank you for your detailed response. However, your answers did not fully address my concerns, and I will maintain my initial rating.
> >
> > Explanation:
> > While I appreciate the idea of integrating GP priors into VAEs, the paper in its current form has two major limitations: (1) no consistent empirical advantage over alternatives and (2) limited theoretical novelty, as it primarily combines existing ideas. I recommend addressing either of these points before resubmitting.

---

### Official Review · Reviewer_Ea83 · 2024-10-30

**Soundness:** 3
**Presentation:** 4
**Contribution:** 4
**Rating:** 8
**Confidence:** 5

**Summary:**

&nbsp;

The authors propose a novel VAE-based Bayesian optimization (VAE-BO) scheme that makes use of a Gaussian process prior VAE to leverage auxiliary covariates in learning a latent space that is better suited for Bayesian optimization. The method is evaluated against several baselines on synthetic data as well as the penalized logP molecule generation benchmark. While the framework is novel and interesting, I have some concerns about the empirical evaluation of the method as well as the reproducibility of the results. In relation to the empirical evaluation, the use of additional covariates that are correlated with the objective would appear to be biased if the baseline methods are not supplied with these covariates. Additionally, I have questions about the discrepancy between the LOL-BO optimization trace reported in the original paper [19] for penalized logP and the trace reported in the current paper. If these issues can be addressed in the rebuttal I will be happy to increase my score.

&nbsp;

**Strengths:**

&nbsp;

1. The work proposes a principled probabilistic framework for VAE-BO leveraging GP prior VAEs to enable conditioning on auxiliary covariates. The scheme is highly general and may be used in conjunction with many disparate VAE-BO architectures.

2. The paper is exceptionally well-written and presented and the scholarship is excellent (an example of which is tracing back the ideas of label guidance to construct discriminative latent spaces to Urtasun et al. 2007).

3. The empirical results are impressive pending the clarifications below as well as the release of the code.

&nbsp;

**Weaknesses:**

&nbsp;

**MAJOR POINTS**

&nbsp;

1. It would be great to perform diagnostic experiments on the GP surrogate fit on the latent space as in Grosnit et al. 2021 [16] in order to validate that improved BO performance is achieved due to a better GP fit on the latent space.

2. On lines 468/469, the authors state, "We augmented the ZINC-250K with five additional covariates: molecular weight, number of hydrogen donors, number of hydrogen acceptors, number of rotatable bonds, and total polar surface area.". Are all baseline methods given access to these covariates? These additional descriptors are highly correlated with the water-octanol partition coefficient logP and so for fair comparison, I would expect that all methods be able to make use of these features in some fashion? It would be great if the authors could clarify exactly how these additional covariates are used for each method.

3. The results reported in Figure 1 of the LOL-BO paper [19] are vastly at odds with the optimization trace reported for LOL-BO in Figure 5b) of the current paper. Why is this the case?

4. The authors do not appear to have released the code for the submission and hence I have some concerns over the reproducibility of the results. This could be supplied as an anonymous GitHub link during the rebuttal phase.

&nbsp;

**MINOR POINTS**

&nbsp;

1. There are some missing capitalizations in the references e.g. "Gaussian" and "Bayesian".

2. When introducing Bayesian optimization, it may be worth citing the originating papers for the method [1, 2] as discussed in [3].

3. The statement that, "Although BO offers an approach for black-box optimisation problems, it does not efficiently scale to high-dimensional data settings" should probably be expanded on in light of recent work [4] which demonstrates that a vanilla GP surrogate where the lengthscale prior is scaled with the dimensionality of the problem can perform effective Bayesian optimization in 100s of dimensions.

4. In terms of the references for applications of VAE-BO, Felton et al. 2020 use a multitask GP surrogate for Bayesian optimization over chemical reaction conditions (as opposed to chemical synthesis) and hence do not make use of a VAE-BO scheme. Additionally, Shields et al. use Bayesian optimization for chemical reaction conditions (as opposed to chemical synthesis) but do not use VAE-BO. Korovina et al. 2020 similarly do not use VAE-BO but rather define an optimal transport kernel directly over molecules. They do however consider chemical synthesis.

5. In Figure 1, the task is articulated as discovering novel drug-like molecules. The penalized logP objective function, however, does not optimize for drug-likeness as noted in e.g. Section 5 of [5]. The penalized logP objective introduced in [6] is misspecified as a metric for drug-likeness since it attempts to maximize logP. For drug-like molecules the logP should however lie within the range of -0.4 to 5.6 according to the commonly-used heuristic Lipinski Rule of 5. As such, I would recommend rephrasing the task to molecule optimization or something comparably generic.

6. In the related work, there has been limited empirical evidence that VAE-BO is beneficial in continuous high-dimensional spaces. In particular, techniques such as random embeddings [7] or SAASBO [8] or more well-known methods. VAE-BO however, has been demonstrated to help in applying Bayesian optimization over structured input spaces such as molecules, images, and biological sequences. As such, it may be worth rephrasing the discussion to focus slightly more on the structured nature of the input spaces as opposed to the dimensionality.

7. The related work does a very good job at covering the majority of VAE-BO methods. Some works that should also be mentioned are [9-12]. Additionally, [13] does not yet appear to be formally published but would be worth mentioning once it is.

8. The citation to Urtasun et al. 2007 shows great scholarship in tracing back the ideas underpinning VAE-BO. Additionally, it would be worth citing Jasper Snoek's PhD thesis [14] which also contained early ideas on label guidance to construct discriminative latent spaces.

9. The decision to use the variable y to represent a high-dimensional observation is somewhat confusing. It may be better to use this variable to represent (noisy) observations of the objective function f.

10. On line 163, it may be worth clarifying the training time complexity is O(N^3).

11. It would be worth citing UMAP [15] given that it is used.

12. Reference [17] should be cited when introducing the Quantitative Estimate of Drug-Likeness (QED) metric.

13. In Figure 5a) it may be worth plotting the log regret to see a clearer distinction between the methods.

14. It would be worth citing t-SNE [18] given that it is used.

15. Reference [20] should be cited when introducing Expected Improvement (EI) as discussed in [3].

16. When discussing Adam in Section C of the appendix it may be worth mentioning that it is an amalgam of the momentum and RMSProp optimizers.

&nbsp;

**REFERENCES**

&nbsp;

[1] Kushner, HJ., [A Versatile Stochastic Model of a Function of Unknown and Time
Varying Form](https://www.sciencedirect.com/science/article/pii/0022247X62900112). Journal of Mathematical Analysis and Applications 5(1):150–167. 1962.

[2] Kushner HJ., [A New Method of Locating the Maximum Point of an Arbitrary Multipeak Curve in the Presence of Noise](https://asmedigitalcollection.asme.org/fluidsengineering/article-abstract/86/1/97/392213/A-New-Method-of-Locating-the-Maximum-Point-of-an?redirectedFrom=fulltext). Journal of Basic Engineering 86(1):97–106. 1964.

[3] Garnett, R., [Bayesian optimization](https://bayesoptbook.com/). Cambridge University Press. 2023.

[4] Hvarfner, C., Hellsten, E.O. and Nardi, L. [Vanilla Bayesian Optimization Performs Great in High Dimensions](https://proceedings.mlr.press/v235/hvarfner24a.html). Proceedings of the 41st International Conference on Machine Learning, in Proceedings of Machine Learning Research 235:20793-20817. 2024.

[5] Griffiths, Ryan-Rhys, Philippe Schwaller, and Alpha A. Lee. [Dataset bias in the natural sciences: a case study in chemical reaction prediction and synthesis design.](https://arxiv.org/abs/2105.02637) arXiv preprint arXiv:2105.02637. 2021.

[6] Gómez-Bombarelli, Rafael, Jennifer N. Wei, David Duvenaud, José Miguel Hernández-Lobato, Benjamín Sánchez-Lengeling, Dennis Sheberla, Jorge Aguilera-Iparraguirre, Timothy D. Hirzel, Ryan P. Adams, and Alán Aspuru-Guzik. [Automatic chemical design using a data-driven continuous representation of molecules.](https://pubs.acs.org/doi/full/10.1021/acscentsci.7b00572) ACS Central Science 4, no. 2, 268-276. 2018.

[7] Wang, Ziyu, Frank Hutter, Masrour Zoghi, David Matheson, and Nando De Feitas. [Bayesian optimization in a billion dimensions via random embeddings.](https://www.jair.org/index.php/jair/article/view/10983) Journal of Artificial Intelligence Research 55, 361-387. 2016.

[8] Eriksson, David, and Martin Jankowiak. [High-dimensional Bayesian optimization with sparse axis-aligned subspaces.](https://proceedings.mlr.press/v161/eriksson21a.html) In Uncertainty in Artificial Intelligence, pp. 493-503. PMLR, 2021.

[9] Verma, Ekansh, Souradip Chakraborty, and Ryan-Rhys Griffiths. [High dimensional Bayesian optimization with invariance.](https://realworldml.github.io/files/cr/paper53.pdf) In ICML Workshop on Adaptive Experimental Design and Active Learning. 2022.

[10] Notin, P., Hernández-Lobato, J.M. and Gal, Y., 2021. [Improving black-box optimization in VAE latent space using decoder uncertainty.](https://openreview.net/pdf?id=F7LYy9FnK2x) Advances in Neural Information Processing Systems, 34, pp.802-814.

[11] Maus, N., Wu, K., Eriksson, D. and Gardner, J., 2023, [Discovering Many Diverse Solutions with Bayesian Optimization.](https://proceedings.mlr.press/v206/maus23a/maus23a.pdf) In International Conference on Artificial Intelligence and Statistics (pp. 1779-1798). PMLR.

[12] Lee, Seunghun, Jaewon Chu, Sihyeon Kim, Juyeon Ko, and Hyunwoo J. Kim. [Advancing Bayesian optimization via learning correlated latent space.](https://proceedings.neurips.cc/paper_files/paper/2023/hash/98e967164ae2f6811b975d686dece3eb-Abstract-Conference.html) Advances in Neural Information Processing Systems 36. 2024.

[13] Chu et al. [Inversion-Based Latent Bayesian Optimization](https://nips.cc/virtual/2024/poster/95013), NeurIPS 2024.

[14] Snoek, Jasper. [Bayesian Optimization and Semiparametric Models with Applications to Assistive Technology.](https://library-archives.canada.ca/eng/services/services-libraries/theses/Pages/item.aspx?idNumber=1033018520) PhD diss., University of Toronto, 2013.

[15] McInnes, Leland, John Healy, Nathaniel Saul, and Lukas Großberger. [UMAP: Uniform Manifold Approximation and Projection.](https://par.nsf.gov/servlets/purl/10104557) Journal of Open Source Software 3, no. 29. 2018.

[16] Grosnit, A., Tutunov, R., Maraval, A.M., Griffiths, R.R., Cowen-Rivers, A.I., Yang, L., Zhu, L., Lyu, W., Chen, Z., Wang, J. and Peters, J., [High-dimensional Bayesian optimisation with variational autoencoders and deep metric learning.](https://arxiv.org/abs/2106.03609) arXiv preprint arXiv:2106.03609. 2021.

[17] Bickerton, G. Richard, Gaia V. Paolini, Jérémy Besnard, Sorel Muresan, and Andrew L. Hopkins. [Quantifying the chemical beauty of drugs.](https://www.nature.com/articles/nchem.1243) Nature Chemistry 4, no. 2 (2012): 90-98.

[18] Van der Maaten, Laurens, and Geoffrey Hinton. [Visualizing data using t-SNE.](https://www.jmlr.org/papers/volume9/vandermaaten08a/vandermaaten08a.pdf?fbcl) Journal of Machine Learning Research 9, no. 11. 2008.

[19] Maus, Natalie, Haydn Jones, Juston Moore, Matt J. Kusner, John Bradshaw, and Jacob Gardner. [Local latent space bayesian optimization over structured inputs.](https://proceedings.neurips.cc/paper_files/paper/2022/hash/ded98d28f82342a39f371c013dfb3058-Abstract-Conference.html) Advances in Neural Information Processing Systems 35, 34505-34518. 2022.

[20] Saltines, VR., One Method of Multiextremum Optimization. Avtomatika i Vychislitel’naya Tekhnika (Automatic Control and Computer Sciences) 5(3):33–38. 1971.

&nbsp;

**Questions:**

&nbsp;

1. My main question relates to how the additional covariates are used for each method. I would be very grateful if the authors could expand on this aspect of the empirical evaluation.

2. For future work the authors may wish to consider the inversion problem [13] namely that under the mapping x -> z -> x' there is typically a reconstruction gap meaning that x is not equal to x'. Enforcing invertibility has been shown in some recent papers to improve VAE-BO performance systematically across architectures. This being said, the contribution is somewhat orthogonal to the contribution of the current work. I believe approaches such as deep metric learning as the authors have compared against are indeed the most appropriate baselines.

&nbsp;

**Details Of Ethics Concerns:**

&nbsp;

No ethical concerns.

&nbsp;

---

> ### Author Response · Authors · 2024-11-24
>
> **"It would be great to perform diagnostic experiments"**
>
> We appreciate the reviewer’s insightful comment. In the Appendices, we have examined the impact of key parameters—specifically, $\beta$ and the choice of kernels—on the quantity of interest. Additionally, we ensured consistency across all experiments by using identical surrogate GP model parameters and the same Expected Improvement acquisition function. The advantages of incorporating GP priors in VAEs are illustrated in Fig. 4 of our manuscript, where our method learns latent representations that exhibit a smooth increase in the target quantity from the lower left to the upper right corner. We are happy to include further details and examples of GP surrogate fits in the latent space in the revised version of our manuscript.
>
> **"My main question relates to how the additional covariates are used for each method... Are all baseline methods given access to these covariates? These additional descriptors are highly correlated with the water-octanol partition coefficient logP"**
>
> We appreciate the reviewer’s feedback because that is related to the core design principle of our proposed method. We would like to highlight the unique capabilities of our model: unlike the baseline methods, our approach is specifically designed to incorporate varying number and type of additional auxiliary covariates via a principled conditional generative model into the high-dimensional BO setting, which we consider a significant contribution. This flexibility enables the utilisation of all available data, including both the auxiliary covariates and the partially observed quantity of interest, to enhance model performance. To the best of our knowledge, none of the previous methods can incorporate additional side information to boost the performance of high-dimensional BO. While we agree that it might be possible to further develop previous methods such that they could incorporate additional covariates, we are not aware of straightforward, principled, and fair ways of modifying the existing methods. Further development of previous methods is, therefore, beyond the scope of our work, and we use the previous methods using their reference implementations, i.e., as they are published.
>
> We also emphasise that the newly introduced auxiliary covariates exhibit varying relationships with the penalised logP. For instance, the number of rotatable bonds influences molecular flexibility, which indirectly impacts logP. High rotatability can penalise drug-likeness and may correlate negatively with penalised logP, further demonstrating the nuanced connections our model can capture.
>
> Additionally, in Fig. 5c of the main manuscript, we evaluate a kernel that considers only the partially observed target quantity of interest. Even under these conditions, our model performs competitively compared to the baselines, showcasing its robustness and offering a compelling alternative to the other VAE BO approaches.
>
> **"The plot traces shown are vastly at odds with the optimization trace reported for LOL-BO"**
>
> The discrepancy stems from the use of different datasets in the penalised logP experiment. As described in our manuscript, we employed the ZINC-250K dataset, originally introduced by Gómez-Bombarelli et al. (2018) and widely used in subsequent benchmarks. This dataset consists of 250,000 drug-like, commercially available molecules from the ZINC database. In contrast, the authors of LOL-BO utilised a smaller snapshot of 10,000 molecules from the GuacaMol training data, resulting in a variation in experimental setups. Of note, we also evaluate and benchmark our proposed method against LOL-BO and other baseline methods on the Expressions dataset, which is exactly the same dataset as used in Maus et al. On the Expressions dataset, our results for LOL-BO as well as for T-LBO (the two best performing baselines) are essentially identical with the results reported in Maus et al., which indicates that our results for the baselines that are obtained using the published reference implementations are computed correctly (or at least are consistent with the previous results reported in the literature).
>
> **"The authors do not appear to have released the code for the submission"**
>
> As stated in the manuscript, we are committed to releasing the source code upon acceptance of the work.
>
> **"For future work the authors may wish to consider the inversion problem."**
>
> We appreciate the reviewer’s suggestion and will certainly take it into consideration for future work.

---

> > ### Author Response · Authors · 2024-11-24
> >
> > **"Minor points"**
> >
> > We greatly value the reviewer’s insightful and detailed comments.
> >
> > - **1., 2., 3., 4., 6., 7., 11., 12., 14., 15.:** We will review and correct the capitalisation in the references and incorporate the additional references suggested by the reviewer in the revised version of the manuscript. Furthermore, we shall make the minor revisions that were requested in the related works section.
> >
> > - **5.:** Figure 1 is intended to illustrate the key aspects of our proposed method through a straightforward example application commonly used as a performance benchmark in the VAE BO literature. Its purpose is to serve as a visual aid, providing clarity on our method within the context of one of the benchmark tasks. We will ensure this is clarified in the manuscript.
> >
> > - **7., 8.:** We sincerely appreciate the reviewer’s kind compliment.
> >
> > - **9.:** Our decision to use $y$ to represent high-dimensional observations relates to our conditional generative model. We follow the standard notation in machine learning and statistics, where inputs (i.e., predictors or covariates) are denoted by $x$ while the outputs (i.e., observations) are denoted by $y$. We understand the potential for confusion and will clarify this further.
> >
> > - **10.:** The time complexity has already been been mentioned in the specified line. We also discuss time complexity in more detail in Section A.1 of the Appendices.
> >
> > - **13.:** The regret is already in the log scale. As described in Section 5.2, the regret is defined as $\log(1 + MSE)$.

---

> > ### Comment · Reviewer_Ea83 · 2024-11-25
> > **Many Thanks to the Authors for their Update**
> >
> > &nbsp;
> >
> > Many thanks to the authors for their update. My response to individual points is provided below:
> >
> > &nbsp;
> >
> > **[P1] Diagnostic experiments for the GP fit in latent space**
> >
> > &nbsp;
> >
> > The experiments I had in mind involved assessing the GP fit via log likelihood on holdout data as in Grosnit et al. 2021. I believe such experiments would help solidify evidence for the mechanism by which the GP prior VAE approach benefits VAE-BO by aiding the construction of a discriminative latent space.
> >
> > &nbsp;
> >
> > **[P2] Auxiliary Covariates**
> >
> > &nbsp;
> >
> > Many thanks for clarifying my question. For competitor methods why can't the additional covariates be incorporated into the featurization for each molecule before it is encoded into the latent space?
> >
> > Many thanks for pointing me towards Figure 5c). In this panel, if I understand correctly, $\Theta_2$ corresponds to the case where additional covariates are not used. In this panel, however, LSO is the only baseline and the performance $\Theta_2$ is not as strong as $\Theta_1$. To show that the authors' method is competitive with existing approaches I believe it would be appropriate to plot $\Theta_2$ in Figures 5a) and 5b) which feature all baselines.
> >
> > I believe this point is crucial for establishing the benefits of the authors' approach as in both experiments, expressions and molecule design, the performance of the method would appear to hinge directly on the use of auxiliary covariates.
> >
> > &nbsp;
> >
> > **[P3] Plot traces are vastly at odds with those reported in the LOL-BO paper**
> >
> > &nbsp;
> >
> > Many thanks for pointing out the difference in the experimental setups! This indeed explains the discrepancy.
> >
> > &nbsp;
> >
> > **[P4] Code release**
> >
> > &nbsp;
> >
> > Unfortunately, it is challenging to evaluate the reproducibility of the results without access to the codebase.
> >
> > &nbsp;
> >
> > In summary, my current outstanding concerns are:
> >
> > 1. A justification for why additional covariates (features) cannot be directly leveraged by other methods.
> > 2. A justification for the authors' method remaining competitive with baselines when additional covariates are not leveraged ($\Theta_2$). This could be achieved by plotting the performance of $\Theta_2$ in the 5a) and 5b) panels for expressions and molecule generation respectively.
> > 3. The absence of code. I believe this is important for establishing reproducibility of the results and without this I don't feel I can upgrade my soundness score and hence my overall assessment.
> >
> > &nbsp;
> >
> > If these points can be addressed I will be only too happy to increase my score and recommend acceptance.
> >
> > &nbsp;

---

> > > ### Author Response · Authors · 2024-11-26
> > >
> > > We thank the reviewer for their careful reading and detailed comments. We hope that the concerns have been satisfactorily addressed. Otherwise, we hope to further engage and address further concerns and suggestions.
> > >
> > > **"A justification for why additional covariates (features) cannot be directly leveraged by other methods."**
> > >
> > > As noted in our previous response, we do not claim that additional covariates cannot be utilised by other methods. However, we are unaware of straightforward, principled, and fair approaches for adapting existing methods to achieve this. Our position aligns with reviewer **naas**, who observed in their review: “*...the case where we have extra 'meta-data' for points in the high-dimensional search space seems perfectly reasonable and impactful (and surprising it hasn't been considered seriously until now).*”
> > >
> > > While it is indeed possible, as the reviewer suggested, to incorporate additional covariates as features into an encoder function, doing so would primarily involve an engineering effort to explore and validate whether a particular encoder architecture and the added covariates enhance high-dimensional BO. In contrast, our contribution lies in proposing a principled framework for integrating additional covariates into high-dimensional BO.
> > >
> > > It is worth noting that since most high-dimensional BO methods rely on VAE-based latent embeddings, our GP prior VAE framework can be incorporated into and applied across all such baseline methods. This provides a principled approach to utilising additional covariates in any VAE-based high-dimensional BO framework. In this manuscript, we demonstrate the effectiveness of our approach with the baseline VAE-BO method, and we see no reason why it could not also prove beneficial for other methods. However, we leave the exploration of these extensions for future work.
> > >
> > > **"A justification for the authors' method remaining competitive with baselines when additional covariates are not leveraged ($\Theta_2$). This could be achieved by plotting the performance of $\Theta_2$ in the 5a) and 5b) panels for expressions and molecule generation respectively."**
> > >
> > > Thank you for reiterating your comment. We believe we now fully understand your perspective. First and foremost, we would like to clarify that our proposed method corresponds to $\Theta_1$, which leverages all available additional covariates through the GP prior VAE model. The other variants, $\Theta_2$ and $\Theta_3$, represent ablations of our model, designed to identify and demonstrate which aspects contribute most significantly to its performance.
> > >
> > > We do not claim that these ablation variants (e.g., $\Theta_2$ and $\Theta_3$) outperform existing baseline models. Regarding your suggestion, we can certainly consolidate Fig. 5b and 5c into a single plot, although they are already presented side by side with the same x- and y-axis ranges. We will incorporate these revisions, along with updates to Fig. 5a (i.e. the Expressions dataset results), into the revised manuscript.
> > >
> > > **"The absence of code. I believe this is important for establishing reproducibility of the results and without this I don't feel I can upgrade my soundness score and hence my overall assessment."**
> > >
> > > We appreciate the reviewer's commitment in reviewing code and establishing reproducibility. We generally officially release code at the time of acceptance but we are happy to provide our code for review purposes via an anonymous link: https://anonymous.4open.science/r/GP-VAE-BO-annon/
> > >
> > > Please note that our official code release will contain more documentation and improved readability.

---

> > > > ### Comment · Reviewer_Ea83 · 2024-11-26
> > > > **Continuing the Discussion**
> > > >
> > > > &nbsp;
> > > >
> > > > **[P1] A justification for why additional covariates (features) cannot be directly leveraged by other methods**
> > > >
> > > > &nbsp;
> > > >
> > > > Common molecular representations (featurizations) include SMILES strings, SELFIES strings, attributed molecular graphs, as well as molecular descriptors. The authors use molecular descriptors (total polar surface area (TPSA), molecular weight, number of hydrogen bonds) as the additional covariates. It is not surprising that incorporating information on the TPSA improves performance as the descriptor is highly correlated with the water-octanol partition coefficient (logP).
> > > >
> > > > A straightforward, principled, and fair approach for existing methods to leverage such descriptors (covariates) would simply be to incorporate them as part of the initial, un-encoded molecular representation x. I agree with the authors that this would necessitate require a VAE architecture to operate on mixed continuous/discrete (heterogeneous) molecular featurizations and I accept that an advantage of the authors' approach is that they can leverage additional covariates without requiring a VAE architecture for heterogeneous data.
> > > >
> > > > Indeed, I believe the comparison between the authors' method and a heterogeneous VAE architecture that explicitly considers molecular descriptors as part of the initial featurization would be interesting. While I appreciate this would be outside the scope of the current work, I believe the authors should not attempt to obfuscate the utility of this comparison. I would suggest rephrasing the relevant sections to emphasize that:
> > > >
> > > > &nbsp;
> > > >
> > > > 1. The improved performance of the authors' method with "auxiliary covariates" is likely due to incorporating molecular descriptors that are correlated with the objective.
> > > > 2. That in principle, the baseline methods could also consider such covariates through the design of a VAE architecture for heterogeneous data [1].
> > > >
> > > > &nbsp;
> > > >
> > > > **[P2] Empirical Performance**
> > > >
> > > > &nbsp;
> > > >
> > > > In light of the codebase release, I understand that the authors are building on the code of Grosnit et al. 2021? Practically, this is incredibly challenging as the codebase of Grosnit et al. is built upon two pre-existing codebases Jin et al. 2018 and Tripp et al. 2020 which themselves contain a mix of ML frameworks (Tensorflow and PyTorch Lightning). In contrast, the codebase of Maus et al. 2022 (LOL-BO) implements a VAE-BO architecture in pure PyTorch. I'm wondering if this codebase would have been easier to work with. I also wonder if the authors' method would perform better if the TuRBO component of LOL-BO was incorporated. I believe it is this observation, that the 56-dimensional latent space is itself high-dimensional and is amenable to trust region optimization, that leads to the strong performance of LOL-BO on the Guacamol dataset.
> > > >
> > > > &nbsp;
> > > >
> > > > **[P3] Code Release**
> > > >
> > > > &nbsp;
> > > >
> > > > Many thanks to the authors for releasing their code. It would be worth including dependency installation instructions. I would also recommend running the code through ChatGPT or Claude to add documentation and/or remove redundancy.
> > > >
> > > > &nbsp;
> > > >
> > > > **__REFERENCES__**
> > > >
> > > > &nbsp;
> > > >
> > > > [1] Ma, C., Tschiatschek, S., Turner, R., Hernández-Lobato, J.M. and Zhang, C., 2020. [VAEM: a deep generative model for heterogeneous mixed type data.](https://proceedings.neurips.cc/paper/2020/file/8171ac2c5544a5cb54ac0f38bf477af4-Paper.pdf) Advances in Neural Information Processing Systems, 33, pp.11237-11247.
> > > >
> > > > &nbsp;

---

> > > > > ### Comment · Reviewer_Ea83 · 2024-11-26
> > > > > **Summary and Score Upgrade**
> > > > >
> > > > > &nbsp;
> > > > >
> > > > > To summarize the discussion, the authors have helped to clarify that the contribution of their method is the ability to leverage auxiliary covariates without requiring the implementation of a VAE architecture for heterogeneous data. I think this is a sufficiently interesting and novel contribution to warrant publication at ICLR.
> > > > >
> > > > > In response to Reviewer t5E1's concern about the the empirical comparison between LOL-BO and the authors' method, I would imagine this is due to the absence of trust region optimization in the 56-dimensional latent space (pending confirmation by the authors that they have omitted TuRBO from their architecture - I couldn't immediately see it in a brief scan of the appendix and codebase.). I believe that:
> > > > >
> > > > > &nbsp;
> > > > >
> > > > > 1. The ablative performance of a single VAE-BO architecture with/without auxiliary covariates is sufficient signal of the efficacy of the authors' approach.
> > > > > 2. The point that has now been clarified, namely that leveraging auxiliary covariates directly in other VAE-BO architectures would require a VAE specifically designed for heterogeneous data is sufficient to mark out the authors' architecture as presenting a novel and potentially advantageous modeling alternative for VAE-BO architectures.
> > > > >
> > > > > &nbsp;
> > > > >
> > > > > Additionally, I have now examined the codebase and can upgrade my soundness score. Overall, I am prepared to champion the paper for acceptance taking it on faith that the authors will implement the additional experiments (diagnostic experiments assessing the log likelihood of the GP fit in the latent space and evaluation of the influence of auxiliary covariates on a LOL-BO style architecture) ahead of the camera-ready deadline.
> > > > >
> > > > > &nbsp;

---

> > > > > > ### Author Response · Authors · 2024-11-28
> > > > > >
> > > > > > We sincerely thank the reviewer for their prompt response and thoughtful reconsideration of their feedback. The additional experiments discussed will be incorporated into the revised manuscript. Furthermore, we wish to clarify that our architecture does not include TuRBO.

---

> > > > > > > ### Comment · Reviewer_Ea83 · 2024-11-28
> > > > > > >
> > > > > > > &nbsp;
> > > > > > >
> > > > > > > Many thanks for the clarification. It would be particularly interesting to run experiments with TuRBO by means of the LOL-BO codebase which would help to isolate the contribution of the auxiliary covariates in the case where there is a method that caters for the high-dimensional latent space.
> > > > > > >
> > > > > > > &nbsp;

---

### Official Review · Reviewer_naas · 2024-11-04

**Soundness:** 3
**Presentation:** 3
**Contribution:** 3
**Rating:** 8
**Confidence:** 3

**Summary:**

Bayesian Optimization is a class of methods for optimizing expensive black box functions $f:\mathbb{R}^d \to \mathbb{R}$ typicality for low dimensional search spaces $d<10$.

In the high dimensional case (e.g. $d>100$) such as the space of images, if we have access to an unlabelled dataset of points in the search space, in this work denoted $y_1,...,y_N \in \mathbb{R}^d$, one may first train a variational autoencoder to map from the high dim search space to a low dim latent space $q:\mathbb{R}^d \to \mathbb{R}^L$ and back $p:\mathbb{R}^L \to \mathbb{R}^d$ where $L<<d$ (simplifying  notation somewhat). Then we simply use $z=q(y)$ and $y=p(z)$ as intermediate translation layers mapping between high and low dimensional spaces. BO is performed in the low dimensional space, modelling a dataset of points $\\{z_i, f(p(z_i))\\}$ with a GP and optimizing the acquisition function over $z\in\mathbb{R}^d$, meanwhile the objective function is evaluated in the high dimensional space $f(p(z))$.

This work considers the case where we have even more data available for some or all of the points in search space denoted $(x_i, y_i) \in \mathcal{X}\times \mathbb{R}^D$ with $\mathcal{X} \subset \mathbb{R}^k$ where $k < 10$. In such a case, we have a regression dataset with low dim inputs and high dim outputs, we may use the GP-VAE architecture. As above with BO, we use the VAE function $z=q(y)$ to convert all the $y_i$ values to low dimensional and now we have a dataset $(x_i, z_i)$ which we can use for normal GP regression, mapping from $x$ to $z$.

**Strengths:**

- __impactful problem__ high dimensional BO and VAE-BO are large problems, and the case where we have extra "meta-data" for points in the high dim search space seems perfectly reasonable and impactful (and surprising it hasn't been considered seriously until now)
- __nice architecture__ the combination of GP-VAE and VAE-BO seems like an intuitive and good choice for such a problem.
- __good benchmarks__ a toy example with MNIST images, mathematical expression tuning and molecule tuning, while the MNIST example  is rather artificial, I felt the molecule example really highlighted the benefit of incorporating covariates.
- __accounting for missing data__ the authors also integrate previous approaches that handle missing data, although this is not novel in this work it is a nice to have and demonstrates broader practicality.

**Weaknesses:**

## Technical Comments
- __Trip 2021 baseline__ is this baseline with weighted retraining or not? It weould be nuice to see a vanilla VAE-BO approach as well as the method of Tripp 2021 with weighted retraining.
- __preference for high valued $y$__ The method of Trip et. al. 2020. starts with a VAE that is a generative model of the whole search space (on a high dim manifold) and after collecting a few fitness values, gradually retrain the VAE to become a generative model of high value parts of the search space, conceptually similar to CMA-ES or a trust region approach. In my view, this method has a nice intuition. In contrast with the above point, I may have misunderstood however it appears as though the proposed ELBO for GP-VAE with missing data does not have a bias for learning high valued $y$.
- __learning without covariates possible failure mode__ when there are no extra covariates beside fitness values $c$, there are two GPs in the latent space,
  - the first GP within the BO algorithm maps from latent points to fitness, modelling $\hat{f}(z): \mathbb{R}^L\to\mathbb{R}$,
  - the second GP maps from fitness back to latent $q(z|c):\mathbb{R}\to\mathbb{R}^L$.

  in a normal BO setting (e.g. VAE encoder and decoder are identity functions) I find this very counter-intuitive, the inverse GP must learn to map from a scalar value $c$ to all the points $\\{z|\hat{f}(z)=c\\}\subset\mathbb{R}^L$, the level set of $c$ for multi-modal function, and this is being modelled by a single uni-modal Gaussian distribution. I have not seen this in the BO literature and it is not immediately obvious why such an approach would help. With _extensive_ retraining the latent space can be remoulded so that the level sets are clustered but this is speculative.

- __limitations__ I may have missed this, but I there does not seem to be much discussion of failure cases and limitations, I have mentioned one above. As with any method that allows to incorporate more data/complexity also allows for more ways to break, if the $x_i$ values are pure noise or if all the optimal $y_i$ points happen to have dramatically different $x_i$ values. The paper does not seems to expose any failure modes or warning for users.

## Minor Comments
- __background__ the proposed method is an intelligent combination of prior methods, and much of section 4 (all of 4.1, 4.2 and parts of 4.3) are describing such prior works and may arguably belong in section 3.
- __section 4.3__ I found this section to be a little bit dense and confusing, adding "yet another" distribution over $x$ (which conditions $z$ which conditions $y$). Although handling missing data is nice and shows practicality, moving this to the appendix as a "bonus feature" and using the space to provide more intuition and justification for the benefit of the main method might be better.

[Tripp et.al. 2020](https://proceedings.neurips.cc/paper_files/paper/2020/file/81e3225c6ad49623167a4309eb4b2e75-Paper.pdf)

**Questions:**

- is it possible to include Trip 2021 as a baseline with and without retraining?
- what is the intuition that means training the GP-VAE would improve the outer BO modelling? Whiles integrating more data can helpat is the justification for the main hypothesis?
- can the autrhors

---

> ### Author Response · Authors · 2024-11-24
>
> **"Does the Tripp et. al, 2020 baseline include weighted retraining?"**
>
> The baseline results for LSO (Tripp et al., 2020) that are reported in our manuscript already incorporates weighted retraining. Following the reviewer's suggestion, we will also include the simpler vanilla VAE-BO in the revised version of our manuscript.
>
> **"Preference for high valued $y$. It appears as though the proposed ELBO for GP-VAE with missing data does not have a bias for learning high valued $y$"**
>
> As detailed in Algorithm 1 of our manuscript, we periodically re-train and fit a conditional generative model, namely GP prior VAE, using the *whole dataset* that consists of the initial dataset as well as the samples collected during the BO steps so far (where covariates may be only partially observed). Contrary to Tripp et al. (2020), our generative model fitting does not have any bias towards high ''fitness values'' (i.e., the quantity that we are trying to optimise). Instead, via periodic generative model training, we are guiding the embeddings of the high-dimensional samples across the whole dataset towards a smooth manifold as specified by the GP prior of our generative model, where the prior is conditional on both the ''fitness values'' as well as all the other auxiliary covariates that may be available in a given application. The BO algorithm is implemented in the learnt latent space that is by design enforced to be well suited for the BO surrogate model, and it is the BO and its acquisition function that eventually learns the preference for high ''fitness values'' (as in classical BO). We will clarify this important distinction in the revised manuscript.
>
> **"Learning without covariates possible failure mode.:."**
>
> We thank the reviewer for their insightful comment. We agree with the reviewer's intuition that a possible failure mode could happen in a normal BO setting (i.e., no embeddings involved) *if* the two GPs were trained simultaneously and *if* the two GPs were enforced to be ''inverse mappings'' of each other. Specifically, our approach does not involve training such a pair of GPs that would involve a mapping to level sets of a multimodal function. Instead, our proposed approach for high-dimensional BO consists of two separate modules that are trained separately: a conditional generative model and the BO surrogate. Training the conditional generative model involves fitting a GP prior from the target quantity (i.e., ''fitness values'') to the latent variables. We hypothesise that leveraging the target quantity (as well as possible additional covariates) as the conditioning variables inherently guides the latent embeddings toward a smooth manifold, which is advantageous for the BO task. The BO surrogate model is then fitted separately using the embeddings from the GP prior VAE for the purposes of identifying latent embeddings and the corresponding high-dimensional data object that optimises the ''fitness values''
>
> **"Discussion of limitations"**
>
> A brief discussion of the limitations of our work has already been included in the Appendices. While our method proposes a novel approach to performing high-dimensional Bayesian optimisation efficiently, it shares several of the limitations of standard VAEs. For example:
> - It can be challenging to model complex (or multi-modal) data.
> - The performance is dependent on the expressiveness of the chosen neural network architecture for the encoder and decoder.
> - The latent variables and their variational approximations are assumed to follow a Gaussian distribution which may not hold true for all datasets.
> - Sensitivity to the hyperparameter values such as dimensionality of the latent space, weight of the KL divergence ($\beta$), minibatch size, etc.
>
> On the other hand, any progress made in learning and deploying VAEs can be incorporated and utilised in our framework as well. Furthermore, in GP prior VAEs, the choice of the auxiliary covariates used for the GP prior needs to be done empirically. Despite these limitations, GP prior VAEs have been successful in various applications and have contributed to advances in generative modelling and unsupervised representation learning. Moreover, in our work, we demonstrate that GP prior VAEs provide competitive performance for high-dimensional BO.
>
> We thank the reviewer for this comment and we will include the above points in the revised version of the manuscript.

---

> > ### Author Response · Authors · 2024-11-24
> >
> > **"What is the intuition that means training the GP-VAE would improve the outer BO modelling? Whiles integrating more data can help - is the justification for the main hypothesis?"**
> >
> > A key limitation of standard VAE-BO is its reliance on an unconditional latent-variable model that lacks guidance from the observed target quantities. In contrast, GP prior VAEs address this by incorporating auxiliary covariates and capturing correlations between data samples — overcoming the assumption in standard VAEs that learned representations are i.i.d.. This enables GP prior VAEs to construct structured, low-dimensional representations that are directly informed by auxiliary covariate information. Additionally, as the reviewer also pointed out, the possibility to easily incorporate a variety of additional application-dependent meta-data via the GP prior VAE is another key strength of our proposed approach. Moreover, GP prior VAEs offer the additional advantage of handling missing values in both high-dimensional observations and auxiliary covariates.

---

### Author Response · Authors · 2024-11-24

We sincerely thank all the reviewers for their thoughtful reading and evaluation of our manuscript.

We will thoroughly revise the manuscript to incorporate the reviewers' comments and suggestions. We believe our responses address the points raised and that the proposed revisions will substantially enhance the quality of the manuscript.

If any of our revisions or responses require further clarification, please do not hesitate to let us know - we would be happy to provide additional details.

We look forward to your feedback.

---

### Meta-Review · Area_Chair_us4x · 2024-12-23

**Metareview:**

The paper tackles the important problem of performing Bayesian Optimization (BO) in high dimensional spaces. To achieve this, the authors employ a GP surrogate with GP prior VAEs. This selection allows for the inclusion of auxiliary covariates for learning the latent space. The reviewers have generally found this method to be impactful and the paper is well-written. They are broadly satisfied with the evaluation provided.

Regarding novelty, on the one hand, there are methods employing VAEs for BOs, and the GP-prior VAE also exists as a model; on the other hand, the overall approach considering VAEs for high-dimensional BO is an interesting and impactful -even if somewhat incremental- idea. Furthermore, the authors discuss the limitations of both standard VAE-BO as well as their method, which makes it clear that there are certain advantages for the proposed method.

**Additional Comments On Reviewer Discussion:**

There has been satisfactory discussion that has helped address various concerns about relation to previous work and empirical performance. This discussion also led one of the reviewers to decide to champion the paper. One of the reviewers still remains unconvinced about the novelty aspect, however overall they don’t see this paper being too far off the threshold.

---

### Decision · Program_Chairs · 2025-01-22

Accept (Poster)